# Suberin Biosynthesis, Assembly, and Regulation

**DOI:** 10.3390/plants11040555

**Published:** 2022-02-19

**Authors:** Kathlyn N. Woolfson, Mina Esfandiari, Mark A. Bernards

**Affiliations:** Department of Biology, Western University, London, ON N6A 5B7, Canada; kwoolfso@uwo.ca (K.N.W.); mesfand2@uwo.ca (M.E.)

**Keywords:** suberin, abscisic acid, transcription factors, CASP proteins, phenylpropanoid metabolism, fatty acid metabolism, macromolecular assembly

## Abstract

Suberin is a specialized cell wall modifying polymer comprising both phenolic-derived and fatty acid-derived monomers, which is deposited in below-ground dermal tissues (epidermis, endodermis, periderm) and above-ground periderm (i.e., bark). Suberized cells are largely impermeable to water and provide a critical protective layer preventing water loss and pathogen infection. The deposition of suberin is part of the skin maturation process of important tuber crops such as potato and can affect storage longevity. Historically, the term “suberin” has been used to describe a polyester of largely aliphatic monomers (fatty acids, ω-hydroxy fatty acids, α,ω-dioic acids, 1-alkanols), hydroxycinnamic acids, and glycerol. However, exhaustive alkaline hydrolysis, which removes esterified aliphatics and phenolics from suberized tissue, reveals a core poly(phenolic) macromolecule, the depolymerization of which yields phenolics not found in the aliphatic polyester. Time course analysis of suberin deposition, at both the transcriptional and metabolite levels, supports a temporal regulation of suberin deposition, with phenolics being polymerized into a poly(phenolic) domain in advance of the bulk of the poly(aliphatics) that characterize suberized cells. In the present review, we summarize the literature describing suberin monomer biosynthesis and speculate on aspects of suberin assembly. In addition, we highlight recent advances in our understanding of how suberization may be regulated, including at the phytohormone, transcription factor, and protein scaffold levels.

## 1. Introduction

Plants have developed effective processes to facilitate their survival, including the production of secondary metabolites [1,2]. The bioactivities of these metabolites involve regulating plant growth, and enabling plants to cope with stressful conditions including biotic threats and environmental hazards [3]. Suberin is a specialized cell wall modifying polymer comprising phenolic and aliphatic compounds derived from phenylpropanoid and fatty acid pathways, respectively, reviewed in [4]. Suberin deposition occurs in two ways: (1) during normal growth in the endodermis and epidermis of roots, bark, seed coats, and specialized organs such as tubers, and (2) in response to wounding stress [5,6]. Suberin is developmentally deposited in the cell walls of root epidermal, exodermal, and endodermal cells in association with the Casparian strip (CS). As a product of secondary metabolism, suberin is regulated and deposited in a tissue-specific manner [7].

The genomic resources of model organisms such as *Arabidopsis thaliana* and the crop potato (*Solanum tuberosum* L.) have provided opportunities to elucidate the function of numerous suberin biosynthetic genes [8,9]. Molecular genetic approaches and mutant analyses have led to the identification and characterization of several genes encoding enzymes involved in suberin biosynthesis including: *β-ketoacyl-CoA synthase* genes *StKCS6* [10], *AtKCS2*, and *AtKCS20* [11,12] encoding enzymes that generate very-long-chain fatty acids, genes encoding cytochrome P450 oxidases required for ω-hydroxy acid biosynthesis, *StCYP86A33* [13,14], *AtCYP86A1* [15], and *AtCYP86B1* [16,17], three *fatty acyl-CoA reductase* genes, *AtFAR1*, *AtFAR4*, and *AtFAR5*, involved in the production of different chain-length primary fatty alcohols [18,19], a glycerol-3-phosphate acyltransferase involved in the synthesis of monoacylglycerol esters encoded by *AtGPAT5* [20,21], aliphatic suberin feruloyl transferases that transfer from feruloyl-CoA to ω-hydroxy acids and fatty alcohols, encoded by *StFHT* and *AtASFT* [17,22,23,24], and genes encoding the ATP-binding cassette G-subfamily transporters StABCG1 [25], AtABCG2, AtABCG6, and AtABCG20 [26]. Since many of the characterized suberin biosynthetic enzymes and their encoding genes exhibit conserved functionality across species, studies in different plant systems are often relevant and applicable to other plant species.

In the last two decades, advances in our understanding of suberin biosynthesis and deposition, and how these processes are controlled, have highlighted regulatory roles for phytohormones, transcription factors (TF), and Casparian strip membrane domain proteins (CASPs). This review provides an overview of the current state of knowledge of suberin monomer biosynthesis, the assembly of monomers into the suberin macromolecule, and the coordination of suberin deposition by phytohormones, TFs and CASPs. An increased fundamental understanding of the role of suberin in response to various stressors, and of the mechanisms that regulate the suberization process, may have important implications for crop improvement efforts, including enhanced tuber storage and resistance to drought stress and pathogen infection [27].

## 2. The Suberin Enigma

Suberin has been differentially described in the literature as a polyester-based aliphatic polymer with associated phenolics (i.e., an integrated model), and as a polymer comprising distinct poly(aliphatic) and poly(phenolic) domains (i.e., a two-domain model). These two views of suberin are based on the knowledge that suberized tissue contains both poly(phenolic) and poly(aliphatic) polymers, but for which the structures are unresolved. The integrated model of suberin structure, originally proposed by Kolattukudy [28], and further refined by Graça [29] depicts a polymer with alternating phenolic and aliphatic polymer layers (Figure 1), inspired by the characteristic lamellar bands observed in TEM micrographs, and supported by numerous lines of evidence, reviewed in [29]. For example, partial depolymerization of suberin yields fragments containing both aliphatic and phenolic compounds, linked via esters of glycerol [29]. In this integrated model, “suberin” refers to a largely poly(aliphatic) polymer with associated phenolics. However, complete removal of the aliphatic components yields a poly(phenolic) rich residue [30] that contains phenolic monomers (e.g., sinapic acid-derived phenolics) not found in the aliphatic portion of the polymer. Additional physical data (solid state ^13^C-NMR [31] and scanning differential calorimetry [30]) supports the two-domain model in which “suberin” refers to a macromolecule with two distinct, but covalently linked domains (Figure 1) [4]. One domain is anchored in the primary cell wall and consists of polymerized phenolic compounds, which is referred to as the suberin poly(phenolic) domain (SPPD) [32,33,34]. The other domain, referred to as the suberin poly(aliphatic) domain (SPAD), spans the space between the cell wall and plasma membrane, and is made up of fatty acid-derived aliphatic constituents, as well as associated phenolics. The two-domain model is further supported by (1) metabolic [35] and gene expression [36,37] data demonstrating a temporal difference in phenolic and aliphatic metabolism during induced suberization, (2) the presence of both phenolic and aliphatic components in ectopically deposited suberin [38], and (3) the failure of suberin deposition in endodermal tissue with compromised phenylpropanoid metabolism [39]. The models are not mutually exclusive, however, and differ mainly in the degree of integration of the phenolic and aliphatic components into “suberin”. Either way, both phenolic and aliphatic metabolism are involved in the biosynthesis and assembly of the suberin in suberized tissues. To complicate matters, recent claims about Casparian strip (CS) composition [40] have questioned the essential composition of suberin, leading the authors to suggest both that (aliphatic) suberin is not essential to CS function, and that poly(phenolics) are not part of “suberin”. However, it can be argued that the data presented in [40] more strongly support a two-domain model of suberin than refute it and reflect a developmental sequence in which phenolics are laid down in the CS in advance of aliphatics. Indeed, more recent data from the same group noted above [39] convincingly demonstrates that when phenolic biosynthesis is impaired during CS formation, suberin lamellae are not deposited in later stages of endodermal development [39]. Altogether, there are many independent lines of evidence supporting a critical role for poly(phenolics) in the overall suberin macromolecular structure.

Regardless of which structural model accurately represents the structure of suberin, the poly(phenolics) consist of polymerized hydroxycinnamic acids and their derivatives, tyramine-derived hydroxycinnamic acid amides (at least in potato), and a small proportion of hydroxycinnamyl alcohols, i.e., monolignols (Figure 2) [32,33,34]. By contrast, the poly(aliphatics) are made up of fatty acid-derived aliphatic constituents including very-long-chain (C24 to C32) 1-alkanols, bifunctional ω-hydroxyalkanoic acids, and α,ω-dioic acids, as well as shorter C18:1 oxidized fatty acids, and non-polymerized (i.e., soluble) associated waxes including alkyl hydroxycinnamate esters, 1-alkanols, fatty acids, and alkanes [41,42,43]. Glycerol and esterified hydroxycinnamic acids are also present amongst the poly(aliphatics) [44,45,46]. The SPAD monomers are cross-linked by glycerol bridges to yield a three-dimensional polyester. While monomers in the SPPD are also cross-linked, these are mediated via inter-unit C-C and ether linkages, rather than ester bonds [4,32]. Evidence supports a likely role for glycerol in linking the SPPD to the SPAD [47,48,49]. Suberized tissues in different species contain varying proportions of these compounds, though ω-hydroxy fatty acids and dioic acids predominate. See for example [29,41,50,51].

## 3. Suberin Biosynthesis and Assembly

Over the last twenty years, many biosynthetic steps required for suberin production have been elucidated. Generally, characterization has focused on steps from two biosynthetic pathways implicated in phenolic and aliphatic monomer production required for the assembly of the respective poly(phenolic) and poly(aliphatic) suberin domains, while novel aspects of linkage and assembly have also been recently elucidated. Table 1 provides a compilation of relevant genes described below.

Total and relative suberin monomer composition varies between developmental stages, plant organs, and species, but is similar enough [50,51] that diverse plants likely express shared pathways required for suberin production. Loss of function mutants have been used to characterize biosynthetic genes in Arabidopsis that translate well to putative potato orthologs in forward and reverse genetics experiments. For example, a key fatty acid ω-hydroxylase gene *AtCYP86A1*, encoding a CYP450 required to produce predominant aliphatic suberin monomers, was characterized in Arabidopsis using a *cyp86a1/horst* mutant [15], and subsequent reverse and forward genetics approaches confirmed the function of its putative potato ortholog *StCYP86A33* [13] including functional complementation of the *cyp86a1/horst* mutant [14].

### 3.1. Biosynthesis and Polymerization of Phenolic Monomers

The chemical composition of the phenolic domain of suberin has not been as thoroughly characterized as that of the aliphatic domain. In potato, which has been most studied in this regard, the SPPD is made up of hydroxycinnamic acids and their derivatives, including hydroxycinnamoyl amides and monolignols (Figure 2), all derived via the phenylpropanoid pathway. This biosynthetic pathway is highly conserved across the plant kingdom and the core pathway is involved in preliminary steps that channel carbon skeletons towards the biosynthesis of many different secondary metabolites, including monomers specific to lignin and the SPPD. The phenylpropanoid biosynthetic pathway is well-established in plants, partly due to its role in lignin biosynthesis. Many steps are common to SPPD biosynthesis.

#### 3.1.1. Phenylpropanoid Metabolism during Suberization

Few studies address the involvement of phenylpropanoid metabolism during suberization. This is partly due to the highly localized and cell-specific deposition of suberin, making it technically challenging to distinguish suberin-specific biosynthesis from that of neighboring tissues. In roots for example, during normal growth and development, phenylpropanoid metabolism associated with CS formation can be overshadowed by the disproportionately greater number of phenolic monomers being synthesized and deposited in lignifying cells in the adjacent stele. Consequently, several distinct and complementary approaches are used to study phenylpropanoid metabolism during suberization: (1) histochemical analyses (primarily to track deposition), (2) molecular approaches using mutant analysis and/or fluorescent-tagged fusion protein expression, and (3) inducible systems such as wound-healing potato tubers. There are several recent and excellent reviews about phenylpropanoid metabolism, for example with emphasis on metabolome formation [87] and transport systems [88]. Herein, we provide an overview of phenylpropanoid metabolism with the intention of providing context for our later discussion about the regulation of suberin biosynthesis and deposition.

Phenylalanine ammonia-lyase (PAL) catalyzes the first committed step of the phenylpropanoid pathway, converting the shikimate pathway-derived amino acid phenylalanine into cinnamate [52]. The next step, *para*-hydroxylation by cinnamic acid 4-hydroxylase (C4H), yields 4-hydroxycinnamic acid, i.e., *p*-coumaric acid [53]. After conversion to its acid-thiol derivative by 4-coumarate-CoA ligase (4CL) [54], *p*-coumaroyl-CoA has several possible metabolic fates. For example, *p*-coumaroyl-CoA can be shunted directly into the flavonoid branch of the phenylpropanoid pathway or serve as a precursor to other branch pathways involving side chain modification (e.g., elongation, conjugation, reduction). An alternative metabolic fate, which, among other end products, channels carbon into suberin monomer formation, involves modification of the *p*-coumaric acid carbon skeleton into other common hydroxycinnamic acids in a series of hydroxylation and methyl-transfer reactions, beginning with the conversion of *p*-coumaroyl-CoA into *p*-coumaroyl-quinate/shikimate via hydroxycinnamoyl-CoA transferase (HCT) [55]. Hydroxylation at the 3′-position by *p*-coumaroyl-quinate/shikimate 3′-hydroxylase (C3′H) yields caffeoyl-quinate/shikimate [56], which is then converted into caffeoyl-CoA by HCT. Methylation of caffeoyl-CoA via caffeoyl-CoA-*O*-methyltransferase (CCoA*O*MT) yields the 4-hydroxy-3-methoxy-substituted hydroxycinnamate structure of ferulic acid [57], which is common to suberin studied in all plant species to date. Ferulate 5-hydroxylase (F5H) [58], in conjunction with caffeic acid *O*-methyltransferase (C*O*MT) [59] in various species yields the 4-hydroxy-3,5-dimethoxy-substituted hydroxycinnamate structure of sinapic acid. *P*-Coumaric acid, caffeic acid, ferulic acid, and sinapic acid (and their derivatives) are all found in the final suberin poly(phenolic) domain, albeit in a species-specific manner.

In potato tuber suberin, hydroxycinnamoyl amides have been identified as part of the SPPD [60]. These derivatives are formed via tyramine hydroxycinnamoyl transferase (THT)-mediated conjugation of tyramine and octopamine, derived from the decarboxylation of tyrosine (tyrosine decarboxylase; TyDC), with hydroxycinnamoyl-CoAs, principally feruloyl-CoA. This pathway is also implicated in the wound- and pathogen-induced biosynthesis of hydroxycinnamic acid amides in leaves of potato [89] and another Solanaceae species, tobacco (*Nicotiana tabacum* L.) [90].

Hydroxycinnamoyl-CoA thioesters can be converted into their corresponding monolignols by cinnamoyl-CoA reductase (CCR) [61] and cinnamyl alcohol dehydrogenase (CAD). Since monolignols make up a small proportion of the SPPD in potato periderm [32], and CCR enzyme activity is lower in suberizing potato tubers relative to lignifying *Pinus taeda* L. cells [91], the monolignol biosynthetic pathway is considered a minor metabolic route for hydroxycinnamoyl-CoAs during tuber suberization [4].

#### 3.1.2. Assembly of the Suberin Poly(phenolic) Domain

Historically, the SPPD of suberized tissues has been considered “lignin-like” due to the presence of phenylpropanoid-derived monomers, including monolignols, cross-linked to the cell wall [28]. However, lignin and the SPPD can be distinguished by their main monomer constituents, especially the relatively high proportion of hydroxycinnamates [4,29]. The assumption that the chemical and structural nature of the SPPD is lignin-like has meant that the details of SPPD assembly are limited and only two main suberin systems have been studied in any significant detail: potato tuber periderm and Arabidopsis CS.

Both the precise localization and mechanics of polymerization are critical for SPPD assembly. Here we focus on the process of polymerization and address recent progress in the coordination of where suberin is deposited in Section 4.4 below. The phenolic domain of suberin was initially proposed to undergo polymerization mediated by peroxidase(s) and H_2_O_2_ over 30 years ago [92], in a process akin to that described for lignification [28,93]. In potato, a suberization-associated anionic peroxidase that preferentially oxidizes hydroxycinnamates over their corresponding alcohols, has been described [62]. The hypothesis that the H_2_O_2_ required for the peroxidase-mediated cross-linking of phenolics is generated by an NAD(P)H-dependent oxidase system is supported by evidence of reactive oxygen species production via oxidative bursts that occur upon tuber wounding [34,63,64]. Similarly, the precise deposition of the CS poly(phenolic) domain was shown to require the spatial production of reactive oxygen species (ROS) via the activation of the respiratory burst oxidase homolog F (RBOHF) NADPH oxidases [65]. Phenolic polymerization in this system requires the localized action of a peroxidase (AtPER64) [66] and a dirigent-like protein (AtESB1) [67]. The involvement of peroxidases, but not laccases, in CS formation in Arabidopsis roots was recently demonstrated using multiple knock-out mutants [68]. Conversely, a salt stress-inducible cationic peroxidase, SlTPX1, has been described in tomato (*Solanum lycopersicum* L.) as playing a role in the polymerization of lignin and phenolic suberin monomers in roots [69]. These species-specific details highlight the differences in suberin poly(phenolic) domain deposition that exist even in closely related plants.

### 3.2. Biosynthesis of Aliphatic Monomers

The SPAD consists primarily of modified fatty acids, primary alcohols, and glycerol (Figure 2). These are ultimately derived from glycolysis and the tricarboxylic acid cycle, which yield acetyl-CoA for plastid-localized fatty acid biosynthesis [94]. Several downstream modifications of 16:0 and 18:0 fatty acid precursors yield final aliphatic suberin monomers characteristic of the SPAD. These include elongation to long (C20, C22) and very-long-chain fatty acids (VLCFA; mostly C24-C32), reduction of long and VLCFAs to primary alcohols, and ω-hydroxylation and further oxidation of ω-hydroxy fatty acids to α,ω-dioic acids. Several enzymes that catalyze these modification steps have been identified and characterized in Arabidopsis and/or potato systems [95] (Table 1).

Acyl activation is typically an initial requirement for downstream fatty acid metabolism and is carried out by long-chain acyl-CoA synthetase (LACS) family enzymes prior to cutin monomer biosynthesis [2,96]. Although no LACSs linked to suberin biosynthesis have been described to date [97], acyl activation of final SPAD monomers may precede linkage of modified aliphatic monomers with glycerol.

Chain elongation and oxidation represent two major fatty acid modification routes that yield the predominant aliphatic suberin monomers. Elongated chains can be reduced to form primary alcohols or decarboxylated to form alkanes, while a large proportion of oxidized fatty acids in the SPAD are short chains (especially C18) that have undergone desaturation instead of elongation. However, some elongated fatty acids are also oxidized to yield VLC ω-hydroxy and α,ω-dicarboxylic acids and are found in the SPAD.

#### 3.2.1. Elongation

Fatty acid elongation is carried out by endoplasmic reticulum membrane-localized fatty acid elongase (FAE) complexes made up of four enzymes [98]. β-ketoacyl-CoA synthases (KCS) are elongase complex enzymes that catalyze the condensation of acyl-CoA with fatty acyl-CoAs, and determine the chain length specificity for each reaction, although single condensing enzymes are able to participate in some consecutive elongation steps [99]. Suberin-related KCSs have been characterized in Arabidopsis and potato. *AtKCS2/DAISY* was first described by Franke et al. [70] as a salt stress-inducible docosanoic acid synthase, since *daisy* mutants produced root suberin that exhibited a concomitant decrease in C22 and C24 VLCFA-based constituent accumulation, with increased C16, C18, and C20 amounts. *AtKCS20* was shown to be functionally redundant with *AtKCS2*, based on similar observations of C22 and C24 reductions in *kcs20* mutants, and a more substantial alteration to root suberin aliphatics in *kcs2 kcs20* double mutants [12]. In potato, *StKCS6* is involved in aliphatic suberin and wax monomer synthesis [10]. *StKCS6* silencing led to a drop in C28 and greater chain lengths, and led to accumulation of C26 and shorter chains, indicating that StKCS6 acts on C26 substrates, but might elongate shorter chains as well.

The KCS-generated β-ketoacyl-CoA is reduced by a β-ketoacyl-CoA reductase to a hydroxyacyl-CoA. Beaudoin et al. [71] described a β-ketoacyl-CoA reductase (*AtKCR1*) that encodes an enzyme catalyzing the first reduction step by the fatty acid elongase complex to yield chain lengths greater than C18 for incorporation into different aliphatic polymers including the SPAD. There have been no potato homologs characterized to date.

The next step involves a 3-hydroxyacyl-CoA dehydrogenase-mediated dehydration of 3-hydroxyacyl-CoA to yield *trans*-2,3-enoyl-CoA. In Arabidopsis, *AtPASTICCINO2* (*AtPAS2*) encodes the third elongase complex enzyme [72]. While suberin-specific aliphatic analysis was not performed in *AtPAS2* characterization, it has demonstrated involvement in synthesizing VLCFA used as precursors for various lipidic compounds, including seed storage triacylglycerols, cuticular waxes, and sphingolipids.

The final enzyme in the fatty acid elongase complex is an enoyl CoA reductase (ECR) that reduces its substrate into the elongated chain with two additional carbons. A gene has been characterized only in Arabidopsis, based on *cer10* mutants. *AtECR* was characterized by Zheng et al. [73] and shown to be required for proper production of cuticular wax, seed triacylglycerols, and sphingolipid production; however, no suberin aliphatics were specifically analyzed.

#### 3.2.2. Oxidation

Oxidation reactions yield modified fatty acids that comprise over half of the SPAD constituents in many plant species, including mid-chain epoxide and hydroxylated octadecanoates, and ω-hydroxyalkanoic acids and their further oxidized α,ω-dioic acid derivatives generated from saturated 16:0 to 24:0 chains as well as 18:1 unsaturated fatty acid. The presence of hydroxylated and dioic VLCFAs suggests that higher chain length products undergo elongation prior to oxidation [4,75]. In potato suberin, ω-hydroxy acids and dicarboxylic acids are predominant and together constitute ca. 65% of the SPAD, with negligible quantities of mid-chain modified fatty acids. In other species, for example cork oak, mid-chain epoxides, and hydroxyalkanoic acids, together comprise >70% of the SPAD [41], while in Arabidopsis, root aliphatic suberin is made up of almost 70% ω-hydroxy acids and dioic acids [100].

Terminal carbon hydroxylations are carried out by cytochrome P450 enzymes belonging to the 86A, 86B, 94A, and 704B subfamilies [101], of which many have been characterized in Arabidopsis and potato. An Arabidopsis CYP86A1 was first enzymatically characterized as a fatty acid ω-hydroxylase by Benveniste et al. [74]. Studies of loss-of-function *Atcyp86a1/horst* and *Atcyp86b1*/*ralph* mutants demonstrated the role of two monooxygenases with varying substrate specificities and functions, where the former yields shorter-chain ω-hydroxy acids (≤C18) and the latter is responsible for the formation of very-long-chain C22-C24 ω-hydroxylated fatty acids in root and seed suberin [15,16]. The *AtCYP86A1* ortholog in potato, *StCYP86A33*, has been characterized in forward and reverse genetics studies, where its silencing led to a reduction in 18:1 and 20:0 ω-hydroxy acids and α,ω-dioic acids in tuber skin and a concomitant increase in 22:0 and 24:0 monomers [13]. *StCYP86A33* expression was found to complement the Arabidopsis *cyp86a1/horst-1* mutant by re-establishing production of oxidized monomers [14]. Complementation of *horst-1* mutants with either *AtCYP86A1* or *StCYP86A33* resulted in an increase in longer-chain distribution than the typically most abundant hydroxylated and dioic 18:1 monomers that is not consistent with RNAi-induced observations [13,14]. This suggests these CYP86As could also use longer chains in addition to their demonstrated shorter-chain (C12-C18) substrates, or this result could be a byproduct of the experimental system [13,14,74].

Oxidation of ω-hydroxyhexadecanoic acid, into its corresponding α,ω-dioic acid, was shown to be carried out by two NADP-dependent oxidoreductases in potato [75,77], though the genes have not been identified. In the sequence of reactions, an ω-hydroxy fatty acid dehydrogenase first oxidizes the ω-carbon of ω-hydroxy fatty acids to produce an ω-oxo fatty acid, which is further oxidized to a dicarboxylic acid by an ω-oxo acid dehydrogenase [75,76]. Only the former appeared to be wound-induced in potato tubers, while the latter demonstrated higher activity than the first enzyme, despite a lack of wound-induced change, suggesting that the rate-limiting step for dicarboxylic acid production is the conversion of ω-hydroxy to ω-oxo acid [75]. Since longer-chain α,ω-dioic acids are present in the SPAD, this suggested pair of enzymes either may not have such a narrow chain length substrate specificity, or there may be other unidentified enzymes responsible for catalyzing these steps with longer chains. Studies in other species suggest that these enzymatic activities are carried out successively by a single monooxygenase. That is, the CYP94 family of monooxygenases have been implicated in catalyzing the formation of cutin and suberin ω-hydroxy fatty acids and dioic acid monomers, by catalyzing ω-hydroxylation of fatty acids. Some enzymes have additionally demonstrated a role in subsequent dicarboxylic acid formation. In vetch, the phytohormone-responsive VsCYP94A1 oxidizes the terminal methyl of C10-C16, C18:1, C18:2, and C18:3 fatty acids [102,103]. Tobacco NtCYP94A5 can oxidize the terminal methyl group of saturated and unsaturated C12-C18 fatty acids into ω-hydroxy fatty acids, except for the C18:0 stearic acid, and the recombinant protein appears to act on 9,10-epoxystearic acid with the highest efficiency. NtCYP94A5 was the first plant enzyme observed to further catalyze the successive oxidation of its preferred substrate into its alcohol, aldehyde and α,ω-diacid counterparts [78]. The Arabidopsis AtCYP94C1 is a wound-responsive enzyme that can hydroxylate saturated, unsaturated and C12-C18 fatty acids, including epoxy-fatty acids. AtCYP94C1 activity exhibited a preference for C12 and C18 chains as substrates, with epoxystearic acid used most predominantly. Heterologous yeast microsome expression experiments demonstrated the ability of AtCYP94A1 to hydroxylate ω-methyl groups and in-chain positions, and to additionally catalyze α,ω-dioic acid formation [79]. While no CYP94s have been characterized to date in potato, Bjelica et al. [14] demonstrated that the gene encoding a putative potato homolog of *NtCYP94A5*, *StCYP94A26*, is expressed in roots and wounded tubers.

#### 3.2.3. Reduction

Fatty acyl-CoA reductases (FARs) catalyze the conversion of fatty acids to primary alkanols for incorporation in the SPAD. Domergue et al. [18] used loss-of-function mutant lines to describe the role of three suberin-related FARs in Arabidopsis; *AtFAR1*, *AtFAR4,* and *AtFAR5*. All are involved in primary alcohol biosynthesis in root suberin and appear to have different saturated chain length substrate specificities. In *far1* mutants, C22 alcohols were reduced in quantity, *far4* mutants showed decreased C20 fatty alcohols and *far5* accumulated lower amounts of C18 alkanols, while heterologous expression in yeast demonstrated a range of chain length specificities from C18-C24 [18]. These fatty alcohol-forming enzymes also contribute toward formation of a large proportion of Arabidopsis root wax alkyl hydroxycinnamates [19,104].

Acyl-activated VLCFAs can also be routed towards the synthesis of waxes that make up a soluble (i.e., unpolymerized) portion of the SPAD [105]. Decarboxylation reduces acyl-CoAs into intermediate aldehydes, and subsequent decarbonylation produces VLC-alkanes. In Arabidopsis, *AtCER1* and *AtCER3* encode core components of a redox-dependent multi-enzyme complex that interacts with electron-transferring cytochrome b_5_ hemoproteins (CYTB_5_s) as cofactors to perform these alkane forming reactions after activation by long-chain acyl CoA synthase, AtLACS1 [2,80].

### 3.3. Esterification, Deposition and Assembly of the Suberin Poly(aliphatic) Domain

The SPAD contains modified fatty acid monomers linked to glycerol, and wax components such as alkyl ferulates that represent the convergence of the main suberin-associated phenolic and aliphatic monomer biosynthetic pathways. Aliphatic monomers such as ω-hydroxy acids and α,ω-dioic acids are linked together by esterification to glycerol [46,49]. The hydrophobic nature of aliphatic suberin constituents requires energetic export of these monomers from the plasma membrane into the lipophobic cell wall. Several plasma membrane-localized ATP-binding cassette (ABC) transporters have been associated with suberin assembly, e.g., [25,26]. This is an aspect of suberin assembly that remains poorly described.

#### 3.3.1. Acyl-CoA Dependent Aliphatic Monomer Esterification

Glycerol 3-phosphate acyltransferases (GPATs) catalyze the transfer of acyl-CoAs to glycerol, to yield monoacylglycerols. GPATs exhibit different regiospecificity, where GPATs capable of catalyzing acylation of the *sn*-2 position of glycerol-3-phosphate represent a land plant-specific lineage of these enzymes [81]. Arabidopsis *gpat5* loss-of-function mutants demonstrate substantial decreases in C20-C24 VLCFA and their ω-hydroxy and dicarboxylic acid derivatives in suberin found in roots and seed coats, and overexpression of *AtGPAT5* led to accumulation of *sn*-2 monoacylglycerols in the wax of Arabidopsis stems [20,21]. These findings support a sequence of biosynthetic events in which monooxygenase-mediated oxidation of a fatty acyl-CoA occurs prior to its linkage with glycerol [20,81]. The wound-inducible *AtGPAT7* is in the same clade as *AtGPAT5*, and its overexpression resulted in the accumulation of suberin monomers [81].

Feruloyl-CoA transferases are involved in the conjugation of ferulic acid with modified fatty acid suberin monomers. Feruloyl transferases have been characterized in Arabidopsis and potato, where feruloyl-CoA acts as an acyl donor in the reaction with an ω-hydroxy fatty acid acceptor to yield ferulate esters. The ferulate esters represent a point of convergence between the two major suberin-related biosynthetic pathways and also may act as a point of connection between suberin domains (see below), as they are proposed to promote linkage of the SPPD and SPAD, as well as between the SPPD and cell wall polysaccharides [5,106,107,108].

In potato periderm, feruloyl transferase *StFHT* is a wound-inducible gene encoding a fatty alcohol/fatty ω-hydroxyacid hydroycinnamoyl acyltransferase that catalyzes the conjugation of ferulate to ω-hydroxyacids and primary alcohols in suberin, and primary alcohols in associated wax. StFHT may have a role in the synthesis of ω-feruloyloxy fatty acids that serve as precursors for ω-feruloyloxy fatty acid glycerol esters [23,24]. In Arabidopsis, aliphatic suberin feruloyl transferase/ω-hydroxyacid hydroxycinnamoyltransferase (AtASFT/HHT) transfers feruloyl-CoA to ω-hydroxy acids with an in vitro preference for 16-hydroxypalmitic acid and also demonstrates transferase activity toward primary fatty alcohols [17,22].

#### 3.3.2. ATP-Binding Cassette (ABC) Transporters

The transfer of suberin monomers, especially aliphatic monomers destined for the SPAD, are reliant in part on half-size ABC transporters and lipid transfer proteins [109]. For example, a rice transporter, RCN1/OsABCG5, is required for root suberization [84], while a pathogen-inducible potato transporter, StABCG1, is involved in the deposition of suberin in tuber skin [25]. RNAi-mediated silencing of StABCG1 led to the reduction of two major C18:1 aliphatic monomers, ω-hydroxy-octadec-9-eneoic acid and its corresponding α,ω-dioic acid along with longer (≥C24) chain ω-hydroxy acids, dicarboxylic acids and fatty alcohols. Less ferulic acid was released in de-polymerized non-polar extracts than in control plants, while feruloyloxy fatty acids and their glycerol esters, and other ferulic acid conjugates had accumulated in the soluble fraction of apolar extracts. These observations indicate that StABCG1 exports major aliphatic monomers including those conjugated to ferulic acid and/or glycerol. More recently, the Arabidopsis homolog of StABCG1, AtABCG1, was shown to contribute to suberin formation in Arabidopsis roots [82]. AtABCG1 was stimulated by its substrates–up to 90% by fatty alcohols and acids with chain length C26-C30 and C24-30, respectively. More importantly, several half-transporters were found to be involved, albeit each with different substrate specificity. StABCG11/WBC11 is the putative potato homolog of an Arabidopsis cuticle lipid exporter, AtABCG11/WBC11 [110]. While StABCG11/WBC11 has not been functionally characterized in potato, its root and tuber-localized expression was shown to be regulated by the suberin-associated transcription factor StNAC103 [83]. Similarly, a putative ABC subfamily G, subgroup WBC/WHITE transporter is highly expressed in suberizing cork oak tissue [7].

Yadav et al. [26] used single, double, and triple *abcg2*, *abcg6,* and *abcg20* loss-of-function mutant Arabidopsis lines to determine the role of these root and seed coat localized ABCG transporters. Aliphatic composition analysis of single mutants highlighted a role for each transporter in suberin production, although they did not have noticeable phenotypic changes. Triple mutants produced suberin with low levels of C20 and C22 saturated fatty acids, C22 alkanol, and C18:1 ω-hydroxy acid and also displayed altered suberin organization and higher water permeability properties in roots and seed coats. The deposition of a suberin polymer in triple mutants, albeit altered, suggests that additional transporters are involved, and highlights the likeliness of substrate specificity along with some redundancy across transporters [26]. Lipid transfer proteins (LTPs) may be a possible supplementary mechanism of suberin monomer export, and while none have been directly linked to suberin deposition, several LTP-encoding genes were upregulated along with known suberin biosynthetic genes and ABCG transporters in an overexpression system studying a transcriptional regulator of suberization [111].

The ABCG transporter required for root suberization in rice, RCN1/OsABCG5, was identified from loss-of-function *reduced culm number 1* (*rcn1*) mutant plants [84]. Relative to wild-type plants, lower quantities of C28 and C30 fatty acids, C16, C28, and C30 ω-hydroxy fatty acids and C16 and C18 dioic acids with a concomitant increase in C24 and C26 ω-hydroxy acids, are found in *rcn1* mutant plants.

#### 3.3.3. Possible Mechanisms for Aliphatic Monomer Polymerization

To date, no definitive mechanism(s) has been described with respect to the assembly of SPAD monomers into a polyester macromolecule. In potato periderm suberin, two key acylglycerol building blocks provide the basis of the SPAD: glycerol-α,ω-diacid-glycerol as the core block, and glycerol-ω-hydroxy acid-ferulic acid to link the SPAD to the SPPD, see [29]. Given the similarity of aliphatic suberin and cutin, it is possible that SPAD monomers undergo an assembly process analogous to that of cutin. Cutin monomers include C16 and C18 ω-hydroxylated acids, mid-chains with epoxy and hydroxyl groups. Primary functional groups can be linked to glycerol at different positions to form 1- and 2-monoacylgylceryl esters [112].

GDSL-lipase/hydrolase family cutin synthases are acyltransferases that mediate the transesterification of major cutin monomers esterified to glycerol. Cutin synthases from tomato fruit (SlCD1/SlCUS1) and Arabidopsis flowers (AtLTL1/AtCUS1) were shown to catalyze a two-step enzymatic polymerization reaction in vitro using a predominant cutin monomer as substrate [113,114,115,116,117,118,119]. First, cutin synthases act on a 2-mono-(10,16-dihydroxyhexadecanoyl)glycerol molecule to generate an acyl-enzyme intermediate while freeing the glycerol moiety, then the intermediate reacts with another cutin monomer to yield a dimer. The polymerization reaction proceeds with fatty acyl groups from more molecules of the same monomer and increasingly larger oligomers as substrates [113,116,118]. However, tomato cutin synthase SlCD1/SlCUS1 produces only linear products from their acylglycerol substrates [115], and suppression of *SlCD1/SlCUS1* expression in RNAi knockdowns and mutants led to a change in esterification of *sn*-1,3 and *sn*-2 positions of glycerol, demonstrating that SlCD1/SlCUS1 acts on primary and secondary hydroxyl groups of its cutin monomer substrates [116]. Together, these findings suggest that additional mechanisms are involved in cutin polymerization, including those responsible for branching and cross-linking of the polymer [115,116]. Due to the compositional similarities between the cutin and suberin polymers, it is feasible that GDSL-esterase/lipase enzymes may be involved in aliphatic suberin assembly, i.e., act as “suberin synthase” proteins analogous to cutin synthases [85]. While there is some overlap between cutin and aliphatic suberin monomers, there are several distinct monomers in each polymer, and therefore such enzymes would require the ability to use suberin-specific acylglycerols as substrates. More recently, the spontaneous self-assembly of covalently linked acyl-glycerol-hydroxycinnamic acid trimers in extracts of potato periderm have been described [120]. These model building blocks of aliphatic suberin formed lamellar-like structures, and open the possibility of a non-enzymatic contribution to the overall structure of the polymer.

### 3.4. The Temporal Deposition of Suberin: Predictions from the Potato Wound Healing Model

Both metabolic [35] and gene expression [36,37] data demonstrate a temporal difference in phenolic and aliphatic metabolism during induced suberization in potato tubers. Using gene expression and chemical analysis data from wound-healing potato tubers, as a guide, the temporal deposition of the phenolic and aliphatic monomers of suberin is predicted to begin with the transport and polymerization of phenolics in the cell wall, followed by aliphatics in the space between the cell wall and plasma membrane (Figure 3). Accordingly, within 24 h post-wounding (hpw), phenolics would begin to accumulate in the cell wall and become cross-linked via a peroxidase-mediated process. The early deposition of phenolics is supported by release of phenolics by thioglycolic acid analysis and monolignols by derivatization followed by reductive cleavage (DFRC) analysis from extractive-free cell walls of wound-induced tubers undergoing suberization [70]. By contrast, it is not until between 48 and 72 hpw that aliphatic suberin monomers/substructures begin to accumulate under the same conditions [119]. A critical step in the overall deposition of suberin, therefore, occurs at the transition between phenolic and aliphatic monomer deposition, since once the aliphatics are in place, their hydrophobic nature would reduce the feasibility of further transport of phenolics into the cell wall. Furthermore, the process cross-linking aliphatics to phenolics remains unclear. One possible mechanism to account for both the transition between phenolic and aliphatic monomer deposition and their cross-linking involves acyl-hydroxycinnamate esters that become cross-linked to the poly(phenolic) matrix. While speculative, this mechanism is supported by the expression of hydroxycinnamoyl transferases such as *StFHT* at this critical time post wounding [37]. Once the phenolic-aliphatic transition is complete, additional aliphatic suberin monomers/substructures would be delivered to the surface of the cell wall where they become cross-linked to aliphatics (Figure 3).

The temporal deposition of suberin in wound-induced potato tubers may serve as a general model for suberin deposition in plant tissue. However, the coordinated deposition of phenolic and aliphatic monomers raises several questions, including how the process is regulated. There is also a limited amount of information about the poly(phenolic) composition of suberin polymers from species other than potato.

## 4. Regulation of Suberization

The differential timing of SPPD and SPAD monomer synthesis and deposition suggests that the enzymes involved in these biosynthetic pathways are controlled by different modes of regulation. The mechanistic details of this differential regulation are not understood, but some aspects of the regulation of suberin biosynthesis have been described (Figure 4). For example, a role for the phytohormone abscisic acid (ABA) has long been implicated as a regulator of suberization, including at the level of gene expression. More recently, transcription factors from different plant species have been demonstrated to control suberin biosynthetic gene expression, and Casparian strip membrane domain proteins (CASPs) have been implicated in the precise cellular deposition of suberin monomers, but especially phenolics.

Phytohormones are involved in complex signaling networks to control physiological events, including the regulation of normal growth and development, and the coordination of abiotic and biotic stress responses. Accordingly, several phytohormones are known to mediate wound-induced signaling and gene expression, e.g., [121]. In the case of suberization, the putative roles of abscisic acid (ABA) [14,24,36,122,123], salicylic acid (SA) [24], jasmonic acid (JA) [122,123], and ethylene [124] have been investigated. However, to date, only ABA has been shown to have a definitive role.

### 4.1. Phytohormone Regulation of Suberization

#### 4.1.1. Abscisic Acid

Abscisic acid is a carotenoid-derived signaling molecule involved in many developmental and stress-related processes in plants, including responses to drought and plant-pathogen interactions [125,126]. Abscisic acid has long been considered to play a regulatory role in potato tuber suberization. Given the established role of ABA in drought and pathogen stress responses, the involvement of ABA in suberization is consistent with the physiological function of suberin in the protection against water loss and infection.

Soliday et al. [127] first posited the likely role of ABA in wound-induced tuber suberization after observing that ABA was released into a solution used to wash cut potatoes. Washing in the first two days after wounding was shown to prevent or delay suberization and the exogenous addition of ABA to washed tuber slices partially reversed this inhibition. Soliday et al. [127] proposed that in the early phase of wound-healing, ABA formation triggers some suberization-inducing factor that lead to the induction of enzymes for suberin biosynthesis. Follow-up work by Cottle and Kolattukudy [45] determined that suberin and its associated waxes increased with ABA treatment. A slight enhancement of ω-hydroxy fatty acid dehydrogenase and phenylalanine ammonia-lyase activity, and a significant increase in suberization-associated peroxidase activity were observed after ABA treatment, linking ABA to the induction of suberization enzymes [45].

De novo ABA biosynthesis is triggered by wounding tubers as shown by an increase in ABA accumulation and the induction of ABA metabolism at the transcriptional level [122,123,128,129]. Lulai et al. [122] provided further evidence of a role for ABA in wound-induced suberin deposition using the ABA biosynthetic inhibitor fluridone (FD), which targets the phytoene desaturase enzyme involved in carotenoid biosynthesis [130], effectively diminishing substrate levels for de novo ABA production [122]. While the impact of FD on suberization was only described using a fluorescence microscopy-based qualitative rating system, the impact of ABA inhibition and exogenous ABA application both appeared to be stronger on the SPAD relative to the SPPD. Endogenous ABA accumulation also promotes potato microtuber dormancy [131], and ABA levels decrease with age in resting tubers. The latter is linked to an age-associated increase in water permeability and loss of wound-healing ability [128].

Several independent lines of evidence have established that de novo ABA biosynthesis is induced upon wounding and promotes the wound suberization processes via upregulation of suberin biosynthetic genes. For example, Woolfson et al. [36] used FD to investigate the role of ABA in the regulation of suberin by inhibiting de novo ABA biosynthesis, with and without the addition of exogenous ABA. Transcript accumulation of key aliphatic suberin-associated genes, including *StCYP86A33*, *StCYP86B12*, *StFAR3*, and *StKCS6* was delayed post-wounding by FD treatment, whereas exogenous ABA application (with or without FD treatment) enhanced transcript accumulation of these same aliphatic suberin associated genes. Similarly, insoluble aliphatic monomer accumulation was reduced in FD-treated tissues. In contrast, FD treatment had no apparent impact on the transcript accumulation of phenolic metabolism genes, while exogenous ABA and the combined FD + ABA treatments slightly increased the accumulation of some polar metabolites [36]. While Kumar et al. [128] demonstrated that the application of exogenous ABA to wound-induced tubers enhanced *PAL1* transcription during phenolic suberin biosynthesis, the overall impact on phenolic metabolism was less clear. Together, these findings suggest a role for ABA in the differential induction of phenolic and aliphatic metabolism during wound-induced suberization, at least in potato tubers [36]. In contrast, Han et al. [132] demonstrated that enzymes involved in polar metabolite synthesis, PAL, cinnamyl-alcohol dehydrogenase (CAD), and peroxidase (PRX), were induced by ABA in wounded kiwi (*Actinidia deliciosa* (Chev.) Liang and Ferguson) fruit tissue. Exogenous ABA application also led to increased content of phenols, flavonoids, alkanes, alkenes, alcohols, alkane acids, olefine acids, esters, glycerides, and tocopherols in wounded tissue. Han et al. [132] concluded that ABA stimulates suberin biosynthesis via activation of PAL, CAD, and POD to accelerate healing in wounded kiwifruit.

Overall, the involvement of ABA in wound-induced and native periderm suberization is evident, but knowledge of its targeted impact on recently characterized biosynthetic steps is limited. The identification of ABA-responsive promoter regions in suberin aliphatic biosynthetic genes, such as cytochrome P450s [14] and feruloyl transferase (*StFHT*; [24]) provide support for the direct control of these genes by ABA. In silico analysis of the putative promoter region of the key potato suberin aliphatic gene *StCYP86A33* led to the identification of several ABA-linked response elements [14]. Regulation of fatty acid ω-hydroxylation is essential during suberin biosynthesis, as more than 55% of aliphatic suberin monomers are oxidized prior to incorporation into the aliphatic suberin domain of the biopolymer [41,119]. The *StFHT* promoter region contains numerous putative hormone- and stress-responsive *cis*-regulatory motifs, including those associated with ABA, jasmonic acid (JA) and salicylic acid (SA) [24]. However, only exogenous ABA application led to the induction of *StFHT* expression in wound-healing tubers [24].

In Arabidopsis, ABA plays a central role in suberin biosynthesis and deposition in drought-induced stress in *Agrobacterium*-induced tumors [133]. Specifically, *Agrobacterium*-induced tumors in Arabidopsis accumulated high amounts of ABA, the ethylene precursor aminocyclopropyl carboxylic acid, osmoprotectants, and formed a suberized periderm-like protective layer. Analysis of gene expression in tumor tissue pointed to a distinct mechanism of drought acclimation in *Agrobacterium*-induced tumors, which differs from that of other tissues, such as leaves or roots. Several suberin related genes regulated by drought and/or ABA showed transcriptional activation in tumors, such as fatty acid ω-hydroxylase (*AtCYP86A1*), phenylalanine-ammonia lyase (*AtPAL1*) and 4-coumarate-CoA ligase (*At4CL2*) as well as a suberin-associated peroxidase and lipid-transfer protein (*AtLTP2*) [133].

Abscisic acid is also involved in the developmental deposition of Arabidopsis root suberin, and likely plays a role in the suberization response to nutrient availability [124]. Barberon et al. [124] established that in the root endodermis, suberization can be enhanced, or suberin can be selectively degraded, depending on the plant’s nutritional status. This plasticity appears to be regulated by ABA (for suberin development) and ethylene (for reversal of suberization) [124]. Using a live marker for suberization, the glycerol-3-phosphate acyltransferase GPAT5 (a key suberin biosynthetic enzyme) [20] and *GPAT5*-driven GUS activity, Barberon et al. [124] demonstrated a match between the pattern of suberin deposition in Arabidopsis roots [40] and *GPAT5* localization via the transcriptional reporter line *GPAT5::mCITRINE-SYP122*. Using this reporter system, it was shown that ABA rapidly induces suberization and led to ectopic deposition in young root parts, as well as in the cortex [124].

#### 4.1.2. Other Phytohormones

Other phytohormones, such as jasmonic acid (JA) and ethylene [120,122], do not seem to be involved in wound-induced suberin biosynthesis in potato tuber, although Barberon et al. [124] provided evidence for ethylene interference during the process of the developmental deposition of Arabidopsis root suberin. That is, ethylene application to developing Arabidopsis roots resulted in apparent degradation of suberin in root endodermal cells [124]; however, aliphatic suberin was not quantitatively assessed. By contrast, inhibition of ethylene biosynthesis had no obvious effect on wound-induced suberin production in potato tubers [123,134]. Similarly, a role for JA in induced suberin biosynthesis remains ambiguous. For example, while the amount of JA increases upon tuber wounding, intracellular levels decline again as ABA and ethylene levels approach their maximum [123]. However, a role for JA in induced suberin biosynthesis cannot be completely ruled out since it has been implicated in the regulation of cell wall damage-induced lignin biosynthesis in Arabidopsis [135], and the suberin poly(phenolic) domain contains monomers in common with lignin. Moreover, as noted above, the suberin-associated gene *StFHT* contains many putative hormone-responsive motifs [24] including those associated with JA and salicylic (SA). However, tubers treated with JA demonstrated no change in, and SA application led to suppression of, *StFHT* expression [24].

### 4.2. Transcription Factors

Transcription factors (TFs) are proteins that bind to specific sequences of their target genes to control transcription, i.e., gene expression. Similar to the role of phytohormones, transcription factors are implicated in the regulation of plant growth and development and in various stress responses and can function within highly coordinated signaling pathways.

Fifty-eight families of TFs, comprising approximately 320,370 members, have been described in 165 plant species [136]. Several major TFs families have demonstrated involvement in biotic and abiotic stress responses [137,138] including: myeloblastosis related (MYB), myelocytomatosis (MYC), NAC domain (NAM, ATAF, and CUC) proteins, WRKY domain proteins, APETALA2/ethylene responsive factor (AP2/ERF), basic helix-loop-helix (bHLH), and basic leucine zipper (bZIP).

In the context of secondary metabolism, TFs have been shown to regulate entire pathways including multiple branches (e.g., *Vitis vinifera* VvMYB5a that controls the phenylpropanoid pathway in grapevine) [139] or subgroups of pathway genes, and under stress-specific conditions (e.g., the wound-, pathogen-, and UV-inducible poplar PtMYB134 that controls the proanthocyanidin branch of phenylpropanoid biosynthesis) [140]. In the context of suberin deposition, MYB, MYC, NAC, and WRKY TFs from Arabidopsis, potato, kiwifruit, cork oak, and apple have been described (Table 2).

**Table 2 plants-11-00555-t002:** Transcription Factors involved in the regulation of suberin biosynthesis and deposition. Transcription factors are summarized according to type and listed in order based on numbering. Multiple entries for a given TF occur when the same TF is described in more than one species or tissue. Only the main tissues in which the TFs have been described are listed, and does not imply that they aren’t functional in other tissues. Similarly, the main signals listed are in reference to the conditions in which a given TF was discovered or tested. Wounding is listed separately from the more generic “abiotic” signal since wounding is a common treatment to initiate suberization. Strictly speaking, no WRKY TFs have been directly shown to be involved in suberization; however, their induction by wounding, and the link between wounding and induced suberization warrant their inclusion.

Transcription Factor		Plant Species	Tissue	Type of Regulation ^1^	Target Pathway ^2^	Signal	Reference
Family	Name						
MYB	MYB1	*Quercus suber*	Cork	+	P	Abiotic	[141]
	MYB4	*Actinidia deliciosa*	Fruit	-	FA	ABA	[142]
	MYB36	*Arabidopsis thaliana*	Root	+	FA, P	Differentiation	[143]
	MYB39	*Arabidopsis thaliana*	Seed coat, root	+	FA, P	Developmental	[85,144]
	MYB41	*Arabidopsis thaliana*	Root	+	FA, P	ABA, abiotic	[38]
	MYB41	*Actinidia deliciosa*	Fruit	+	FA	ABA, developmental	[145]
	MYB74	*Solanum tuberosum*	Wound periderm	+	FA	Wounding	[146]
	MYB93	*Malus* × *domestica*	Fruit skin	+	FA	Developmental	[111]
	MYB102	*Solanum tuberosum*	Wound periderm	+	FA	Wounding	[146]
	MYB107	*Arabidopsis thaliana*	Seed coat	+	FA, P	Developmental	[85]
	MYB107	*Actinidia deliciosa*	Shoot, leaf	+	FA	ABA	[145]
×MYC	MYC2	*Actinidia deliciosa*	Root, shoot, leaf	+	FA	ABA	[145]
NAC	ANAC046	*Arabidopsis thaliana*	Root, floral bud, fruit, leaf, wounded leaf	+	FA	Wounding, developmental, senescence	[147]
	NAC103	*Solanum tuberosum*	Wound periderm, leaf, periderm, root	-	FA	ABA, wounding, developmental	[83]
WRKY	WRKY1	*Solanum tuberosum*	Stem	+	P	Biotic/abiotic	[148]

^1^ + = positive; - = negative. ^2^ P = phenylpropanoid; FA = fatty acid.

#### 4.2.1. MYB and MYC Transcription Factors

Several transcription factors have been described as regulators of suberin biosynthesis and deposition and are also important components of ABA signaling in response to abiotic stresses in plants. For example, Kosma et al. [38] confirmed that the transcription factor AtMYB41 is a positive regulator of suberin biosynthesis and deposition under abiotic stress conditions. Overexpression of *AtMYB41* in leaves of *Nicotiana benthamiana* resulted in the ectopic accumulation of transcripts for aliphatic suberin biosynthesis genes (e.g., *AtCYP86A1*) and related metabolites (ω-hydroxy fatty acid and dicarboxylic acids). Overexpression of *AtMYB41* also led to increased phenylpropanoid and lignin biosynthetic gene transcripts and production of some metabolites like monolignols in leaves. Furthermore, the ectopic deposition of suberin in leaf epidermal and mesophyll cells resulting from overexpression of *AtMYB41* resembled that typically observed in roots and native and wound potato tuber periderm [38]. Similarly, overexpression of *AtMYB92* in *N. benthamiana* resulted in a significant increase in the accumulation of suberin-related aliphatics [149].

*MdMYB93* was identified by Legay et al. [150] in russeted apples (*Malus* × *domestica*) with high suberin deposition in fruit skins, via comparative transcriptomic analysis with non-russeted varieties. The authors demonstrated a correlation between *MdMYB93* expression and putative apple homologs of *AtCYP86A1*, *AtGPAT5*, and *At**CYP86B* [150]. Subsequently, Legay et al. [111] used a heterologous *N. benthamiana* expression system to transiently overexpress *MdMYB93* and measure its impacts on suberin related metabolism and transcriptome-wide gene expression. *MdMYB93* expression in agroinfiltrated *N. benthamiana* leaves was accompanied by increased accumulation of suberin biosynthetic gene transcripts including those involved in cell wall development, lipid, and phenylpropanoid metabolism and ABCG family transporters. Overexpression of *MdMYB93* also changed the composition of phenolic metabolites in *N. benthamiana* leaves [111], though their potential role in SPPD formation was not investigated.

Transcriptomic studies of suberization in wounded cuticle-deficient tomato (*Solanum lycopersicum*) and russeted apple (*Malus* × *domestica*) fruit skins demonstrated the upregulation of suberin-related genes in suberized skin tissues [85]. Comparative transcriptome co-expression analyses using additional plant species and tissues revealed a conserved gene expression signature of 26 genes for suberin monomer synthesis and polymer assembly among angiosperms. Co-expression analyses led to the identification of putative orthologs in a suberin-linked MYB clade, including tomato *SlMYB93*, apple *MdMYB53*, grape *VvMYB107*, potato *StMYB93*, rice *OsMYB93*, and Arabidopsis *AtMYB107* and *AtMYB9* [85]. Lashbrooke et al. [85] confirmed the role of *AtMYB107* and *AtMYB9* in suberin biosynthesis and assembly through the analysis of loss-of-function mutants, in which they observed a significant reduction in suberin phenolic and aliphatic monomer components. The authors concluded that AtMYB9 and AtMYB107 coordinate the transcriptional regulation of phenolic and aliphatic monomer biosynthesis and deposition in suberizing tissue.

A combination of targeted metabolomics and transcript profiling in wounded tubers, and heterologous expression in *N. benthamiana* led to the identification and initial characterization of two additional potato TFs, StMYB102 and StMYB74, that regulate wound suberization processes [85,146]. The accumulation of significant amounts of aliphatic suberin monomers in leaves of *N. benthamiana* expressing either StMYB102 or StMYB74 supports a role for these two TFs in the regulation of aliphatic suberin deposition. Since phenolic deposition is required for normal suberin lamella deposition [39], and heterologous expression of *StMYB102* and *StMYB74* in *N. benthamiana* leaves resulted in the deposition of suberin lamellae between the plasma membrane and cell wall, StMYB102 and StMYB74 are also implicated in the regulation of suberin phenolic deposition. Differential expression of *StMYB102* and *StMYB74*, and single nucleotide polymorphisms in *StMYB102*, all correlate with the amount of wound-suberin deposited by different cultivars [146]. These data provide important insight into the basis of quantitative differences in the amount of suberin deposited in different potato cultivars.

Cohen et al. [144] described SUBERMAN (SUB), a MYB-type transcription factor (i.e., AtMYB39) associated with the deposition of suberin monomers in the root endodermis layer that acts as a positive regulator of suberin deposition during normal development. Transient expression of *AtMYB39* in *N. benthamiana* leaves resulted in the induction of heterologous suberin genes, the accumulation of major suberin monomers, and the deposition of suberin-like lamellae. Indeed, SUB activated the promoters of genes involved in suberin deposition and therefore could control the expression of suberin biosynthesis genes. Analysis of gene expression profiles provided insight into the influence of *SUB* on root endodermis suberization during natural development. Generally, SUB affects transcriptional networks linked with suberin deposition, including the biosynthesis of phenylpropanoids, lignin (i.e., poly(phenolic) domain), very-long-chain-fatty-acid (VLCFA)-CoA, cuticular lipid components, root transport activities, hormone signaling, and cell wall regulation. SUB regulation of suberin biosynthesis and deposition appears to be specific to root endodermis, but does not affect CS formation [144]. The latter point supports the notion that CS formation and suberin lamellae deposition are distinct processes [124], but does not rule out a role for both phenolic and aliphatic monomer deposition in CS, as measured for maize [151] and soybean [152].

*QsMYB1* is a MYB family member described from cork oak (*Quercus suber*) and shown to be involved in the regulation of phenylpropanoid and lignin pathways expressed in organs and tissues that undergo secondary growth [141]. *QsMYB1* expression is moderated by an alternative splicing mechanism that results in two different transcripts: *QsMYB1.1* and *QsMYB1.2*. Each transcript was differentially expressed, depending on the nature of the abiotic stress applied to the tissue. For example, accumulation of *QsMYB1.1* transcripts were downregulated slowly at high temperatures, while *QsMYB1.2* was temporarily upregulated in response to drought stress [153]. Capote et al. [154] developed a ChIP-Seq strategy for analysis of the DNA targets of *QsMYB1* across the cork oak genome, and determined that several genes were targeted by QsMYB1, including those encoding enzymes for monolignol and phenolic suberin component biosynthesis. Additionally, several members of the ABCG gene family and lipid-transfer proteins (LTPs) were regulated by QsMYB1, which signifies the important role of *QsMYB1* in the regulation of lipid transport and suberin formation across the cellular membrane [154].

In kiwifruit, five TFs, AchnMYC2, AchnMYB4, AchnMYB41, AchnMYB107, and AchnABF2, which regulate expression of *Ac**hnCYP86A1* [142,145], *AchnFHT* [155], and *AchnFAR* [142,156] have been identified. More specifically, AchnMYC2, AchnMYB41, and AchnMYB107 individually interacted with the *AchnCYP86A1* promoter to activate gene expression [145], and heterologous expression in *N. benthamiana* resulted in the accumulation of ω-hydroxy acids, α,ω-diacids, fatty acids and primary alcohols. The increase in primary alcohols was later shown to correlate with increased *AchnFAR* expression [156]. Similarly, yeast 1-hybrid and dual-luciferase analyses demonstrated that AchnABF2, AchnMYB41, and AchnMYB107 activated the *AchnFHT* promoter, while AchnMYB4 acted as a repressor [155]. The authors concluded that the activation of suberin assembly, involving AchnFHT, was fine-tuned through both repression by AchnMYB4 and promotion by AchnABF2, AchnMYB41, and AchnMYB107.

#### 4.2.2. NAC Transcription Factors

The NAC family constitutes one of the largest transcription factor families in land plants and is represented by 110 genes in potato [157]. NAC transcription factors are involved in various plant functions from development, lateral root formation and auxin signaling [158] to secondary cell wall biosynthesis [159], and stress response regulation [160]. In potato, a large proportion of NAC genes are sensitive to abiotic stresses [83,161]. StNAC103 was recently described by Verdaguer et al. [83] as a wound- and ABA-inducible negative regulator of aliphatic suberin and wax accumulation. Its promoter drove GUS expression in native and wound periderm phellem cells undergoing suberization, and also in tissues that do not produce suberin, namely lateral root primordia and root apical meristems. Its RNAi-mediated gene silencing led to the significant upregulation of genes with predicted or known involvement in different branches of suberin-related metabolism: *StCYP86A33*, *StKAR*, *StFHT*, and *StABCG11*/*StWBC11*. Although TF binding activities were not investigated, the impact of *StNAC103*-silencing on these genes was consistent with an increase in their substrates, such as alkylferulate wax components, ω-hydroxy acids, primary alcohols, and alkanes. The observed transcriptional repression activities of StNAC103 likely control suberin synthesis during wound-healing on a fine scale, and probably function to prevent premature suberization in certain root localizations in vivo [83]. In Arabidopsis, AtANAC046 facilitates the deposition of suberin in roots, particularly in the endodermis of primary roots and the periderm of secondary roots, and in leaves in response to wounding [147]. A comparison between overexpression (OE) and wild-type (WT) lines indicated that the expression of suberin biosynthesis genes was higher in the roots and leaves of OE lines compared to the WT. Moreover, the amount of aliphatic suberin content (fatty acids, specifically very-long-chain fatty acids (VLCFA) of C24 and C26 as well as sterols) in roots of Arabidopsis was doubled in OE lines relative to WT. The authors proposed that AtANAC046 primarily mediates aliphatic suberin biosynthesis in Arabidopsis roots [147].

#### 4.2.3. WRKY Transcription Factors

While a role for WRKY TFs in suberization has not been explicitly demonstrated, their ability to activate transcription of phenylpropanoid genes involved in SPPD monomer synthesis, like *StTHT* [148], offer insight into a possible regulatory mechanism for stress-induced phenolic suberin production. Indeed, WRKY-domain TFs are involved in many plant processes including responses to both biotic and abiotic stresses [148,162,163,164,165,166]. For example, Yogendra et al. [148] characterized StWRKY1 as a regulator of pathogen-induced hydroxycinnamic acid amide deposition into secondary cell walls of aerial potato tissues. The authors used metabolomic comparisons between pathogen-inoculated late blight (*Phytophthora infestans*)-resistant and -susceptible potato cultivars to elucidate phenylpropanoid metabolites linked to resistance, and identified the genes required for their biosynthesis, *4-coumarate:CoA ligase* (*St4CL*) and *tyramine hydroxycinnamoyl transferase* (*StTHT*). StWRKY1 was shown to physically interact with target DNA, and its function was further validated using a gene silencing approach. StWRKY1 binds to the promoter region of its target phenylpropanoid biosynthetic genes, *St4CL* and *StTHT*, to activate transcription. Gene silencing of *StWRKY1* caused a significant decrease in hydroxycinnamic acid amide abundance and a severe reduction in resistance to the late blight causing pathogen *P. infestans* [148]. These results revealed a role for StWRKY1 in the regulation of downstream genes during hydroxycinnamic acid amides synthesis in stress-induced metabolism, and implicate it in the formation of wound suberin.

### 4.3. Interaction between Phytohormones and Transcription Factors

In the context of suberin biosynthesis, linkage, and deposition, a picture is emerging in which several TFs coordinately regulate suberin-specific gene expression under various developmental and stress conditions, in conjunction with specific phytohormones (especially ABA). Recent evidence supports the interaction between the TF-mediated developmental programs and stress hormone-mediated pathways in the regulation of Arabidopsis root endodermis suberization [143,167]. For example, the SHORT-ROOT (AtSHR) transcription factor, and AtMYB36 are organized into a sub-network that regulates CS formation [168,169]. Several of the genes regulated by AtSHR and AtMYB36 include other MYB transcription factor genes including *MYB9* [85], *MYB107* [85,170], *MYB41* [38], *MYB39* [144,171], and *MYB93* [111]. Wang et al. [171] illustrated that *MYB39* and *MYB93* are co-regulated by *SHR* and *MYB36*. They concluded that *MYB39* could function as a potential downstream hub connecting the *SHR* regulatory network and ABA pathways to promote suberization [143]. Similarly, Shukla et al. [167] recently demonstrated a direct role for a small suite of MYB TFs, namely AtMYB41, AtMYB53, AtMYB92, and AtMYB93, in Arabidopsis CS suberization in response to ABA and the SGN3/CIF pathway (see Section 4.4, below).

In kiwifruit, expression of several TFs has been linked to ABA [142,145,155,156]. Specifically, *AchnMYB41*, *AchnMYB107*, and *AchnMYC2* expression was enhanced in response to ABA, whereas FD suppressed the expression of genes encoding these TFs [156]. The overexpression of these TFs in *N. benthamiana* leaves led to the upregulation of aliphatic suberin synthesis genes and an increase in the amounts of primary alcohols, α,ω-diacids, ω-hydroxyacids, and fatty acids [156]. In contrast, AcnhMYB4 was found to suppress the expression of aliphatic suberin biosynthetic genes *AchnCYP86A1* and *AchnFAR* and reduce accumulation of ω-hydroxyacids and primary alcohols in wounded kiwifruit and in a *N. benthamiana* expression system [142].

Activator and suppressor TFs appear to work in concert to fine-tune regulation, either for both phenylpropanoid and fatty acid metabolic pathways, or solely aliphatic suberin deposition. The majority of defined suberin-related TFs are transcriptional activators, including AtMYB9, AtMYB39, AtMYB41, AtMYB92, AtMYB107, and AtANAC046 in *Arabidopsis thaliana*, StMYB74 and StMYB102 in potato, MdMYB93 in apple, QsMYB1 in *Quercus suber* and AchnABF2, AchnMYB41, AchnMYB107, and AchnMYC2 in kiwifruit (summarized in Table 2). In contrast, StNAC103 in potato and AchnMYB4 in kiwifruit repress the biosynthesis of aliphatic suberin, as well as the cross-linkage of aliphatics with ferulic acid, and its export. Interestingly, *StNAC103* gene expression is induced by ABA, whereas *AchnMYB4* is inhibited by exogenous ABA application [83,156]. AtMYB107 and AtMYB9 act together to regulate both SPPD and SPAD metabolism [85].

TFs function within highly complex regulatory networks, which may involve phytohormone-signaling, to control suberin-related metabolism and transport in different species, during normal development and under stress conditions. Further research is required to fully elucidate the regulatory oversight of differential suberin-related pathway induction during the wound response, including the timing of biosynthesis, the spatial organization of deposited suberin, and the fine-tuning that is likely required to harmonize this process.

### 4.4. A Role for Casparian Strip Membrane Domain Proteins

Roots absorb nutrients from the soil and transport them through the epidermis, cortex and endodermis layers to reach to the vasculature (stele). Transport of nutrients to the stele takes place via two major pathways. The first is the apoplastic pathway, in which solutes diffuse freely through the apoplast. The second is the symplastic pathway, where solutes move cell-to-cell through the plasmodesmata [172]. The endodermis is the innermost cell layer that surrounds the vasculature [94]. During differentiation, the endodermis undergoes specialized cell wall modifications including the deposition of lignin-like poly(phenolic) and aliphatic suberin polymers in a continuous band in their radial walls, forming the Casparian strip (CS) in what is referred to as a State I endodermis [151]. After formation and maturation of the CS, the endodermis undergoes a second step of differentiation, forming a State II endodermis in which lamellar suberin (primarily aliphatic suberin monomers) is deposited between the primary cell wall and the plasma membrane and completely covers the cell surface [173]. Therefore, it seems that the poly(phenolic) and poly(aliphatic) suberin domains of the endodermis are regulated separately and in a spatiotemporal manner. Initial deposition of lignin-like poly(phenolic) components delineates the formation of the CS, building a ring-like structure that connects the radial walls of each endodermal cell to its neighboring cells, to form a barrier to diffusion through the apoplast [40,65,174]. In Arabidopsis mutants defective in aliphatic suberin deposition, the presence of the poly(phenolic) domain of suberin alone appears to be sufficient to form a functional CS [40]. This does not preclude aliphatic suberin as an integral component of the CS, however, and clear chemical evidence has been presented in support of significant levels of aliphatic suberin in State I endodermal tissues of some species, including maize [151] and soybean [152]. At early stages of endodermal differentiation in Arabidopsis (i.e., the State I-II transition), a patchy distribution of aliphatic suberin precedes complete suberization of endodermal cells, while fully differentiated mature endodermal cells have a continuous suberization zone [172].

Roppolo et al. [175] identified a family of transmembrane proteins called Casparian strip membrane domain proteins (CASPs). CASP deposition predicts the site of CS formation in the endodermal cell wall [175]. CASP1-5 are members of this family in Arabidopsis, and their aggregation marks a membrane domain that attaches to the cell wall, excludes most other plasma membrane proteins and delineates the location of the CS [65,175,176]. CASP proteins appear to form a scaffold that acts as a platform for regulation and deposition of the CS poly(phenolic) domain [67,175]. The precise deposition of the CS poly(phenolic) domain requires a signaling pathway involving a cytoplasmic receptor-like kinase SCHENGEN1 (AtSGN1) and the receptor SCHENGEN3 (AtSGN3), which control the spatial production of reactive oxygen species (ROS) via the activation of the respiratory burst oxidase homolog F (RBOHF) NADPH oxidases [65,177,178,179]. Phenolic polymerization requires the localized action of a peroxidase (AtPER64) [66] and a dirigent-like protein (AtESB1) [67]. Thus, the SGN3 signaling pathway operates in parallel with a network of transcriptional factors involving SHR-SCR and AtMYB36 that controls endodermal differentiation [180,181].

CASPs are found in many plant species [86,182], where their function may be conserved. In potato, two orthologs of Casparian strip associated Arabidopsis CASPs, *StCASP1-like/StCASP8* and *StCASP1B2-like/StCASP9*, are expressed during tuber periderm maturation (i.e., in suberizing potato skin) along with a cutin synthase-like GDSL lipase/esterase gene (*StGDSL-like*) [86]. It is tempting to speculate that CASPs and GDSL-like proteins may also be involved in coordinating the deposition of suberin in response to wounding. For example, Woolfson [37] demonstrated that these two putative CASP genes identified in tuber phellem were also upregulated by wounding in potato tubers, along with other suberin-associated genes [37]. Such a role for CASP genes outside CS formation is not unprecedented as AtCASPL1B2 (a putative homolog of StCASP1B2-like/StCASP9) was downregulated in *myb9 myb107* mutant seed coats [85]. *StCASP1-like/StCASP8* was expressed in the same temporal pattern as genes required for SPAD assembly and coincided with the upregulation of aliphatic monomer biosynthesis genes, along with the putative GDSL-type gene *StGDSL.7* [37]. This suggests that StCASP1-like/StCASP8 and StGDSL.7 may act to regulate aliphatic monomer incorporation into the SPAD in potato at a later time than when genes are first expressed and soluble monomers accumulate. Moreover, StCASP1-like/StCASP8 could act as a mediator between SPPD and SPAD assembly activities, possibly by orchestrating the linkage of the two domains. Additionally, *StCASP1B2-like/StCASP9* and another putative GDSL-type gene, *StGDSL.6*, followed temporal expression patterns similar to aliphatic metabolism genes after wounding, and thus could be candidates for involvement in SPAD assembly [37].

New candidate proteins involved in CS formation were identified during characterization of other AtMYB36-regulated genes [183] in Arabidopsis, including a gene encoding a copper-containing protein Uclacyanin1 (UCC1) that localizes to the CS. UCC1 constitutes a central CS nanodomain while other CS-located proteins are accumulated at the periphery of the CS. The loss-of-function of two uclacyanins (UCC1 and UCC2) decreased poly(phenolic) deposition, specifically in the central CS nanodomain and led to endodermal permeability and altered mineral nutrient homeostasis [183]. This result suggests that CS biosynthetic machinery involved in the polymerization of phenolics is nano-compartmentalized.

CASPs are essential for the precise localization of the CS by forming a scaffold domain (i.e., Casparian strip domain) within the plasma membrane that predicts the formation of the CS. Indeed, the Casparian strip domain provides a protein platform that allows localization of various enzymes like peroxidases, reactive-oxygen species-producing enzymes, transporters or combinations of these. Despite new understanding of the underlying mechanisms leading to CS initialization, many questions remain about the complex regulatory network underlying CS formation. Similarly, a more general role for CASPs in delineating sites for suberin deposition may exist, considering their wound-inducibility in potato tubers. Further studies on the identification and characterization of the CASPs and their associated proteins will offer more insights into the molecular mechanisms that determine the development, early differentiation, and CS formation in endodermal cells, and suberin deposition in general.

## 5. Conclusions

Timely and effective suberization is vital to prevent water loss and infection, as well as healing wounds incurred during harvest and post-harvest handling and storage. This review provides insight into the molecular-genetic pathways leading to the biosynthesis and regulation of suberin deposition. There remains a need to elucidate the differential and temporal regulation of phenolic and aliphatic metabolic pathways required for suberin development. Recent investigations into the regulatory oversight of suberization have revealed the involvement of phytohormones including ABA, transcription factors that interact with ABA and biosynthetic genes, and novel candidates that may coordinate suberin domain assembly and linkage, such as CASPs. Collectively, these findings advance our fundamental understanding of what orchestrates the suberization process, and the potential to improve valuable agricultural products through the targeted engineering of crops with improved stress resistance traits, such as enhanced drought and pathogen resistance, or in identifying molecular markers to assist in the selection of crops with improved agronomic traits and post-harvest storability.

## Figures and Tables

**Figure 1 plants-11-00555-f001:**
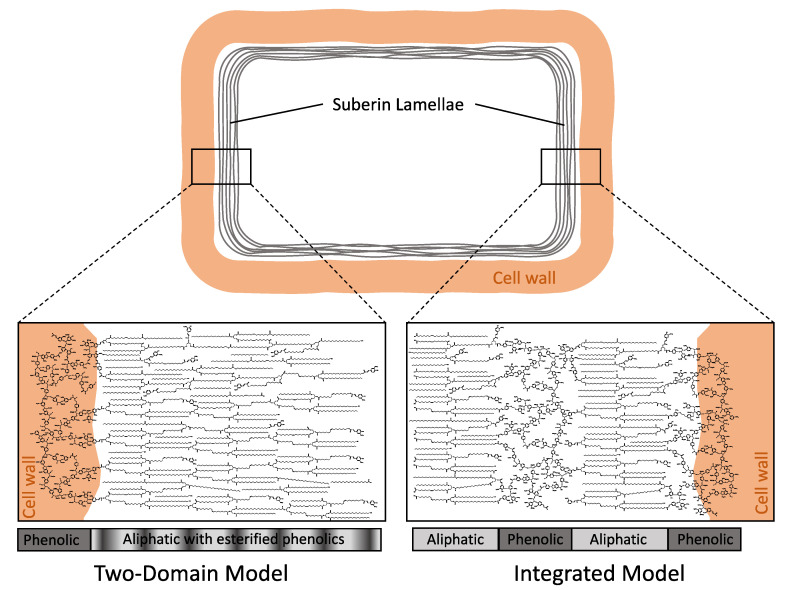
Comparison of two models of suberin structure. Two models of suberin structure are presented: a two-domain model based on [4] and an integrated model based on [29]. In each case, both phenolic and aliphatic polymers are depicted. The poly(phenolics) are shown as cross-linked by C-C and C-O-C bonds, while the poly(aliphatics) are shown as polyesters cross-linked via glycerol. In the two-domain model, the poly(phenolic) domain is envisioned as integrated into the cell wall, and esterified to the poly(aliphatic) domain via glycerol esters. The characteristic lamellae of suberized cells are proposed to arise from variation in electron density moving through less dense hydrocarbon and more dense areas rich in ester linkages and phenolics. By contrast, the integrated model is shown as repeating layers of poly(phenolic) and poly(aliphatic) components, that give rise to the characteristic lamellae. The degree to which the integrated model is embedded into the cell wall, or how tightly the two polymer types are cross-linked, remains unknown. Several lines of evidence support both models; further targeted experimentation will be required to determine whether one or both exist, or indeed a different arrangement comprises suberin.

**Figure 2 plants-11-00555-f002:**
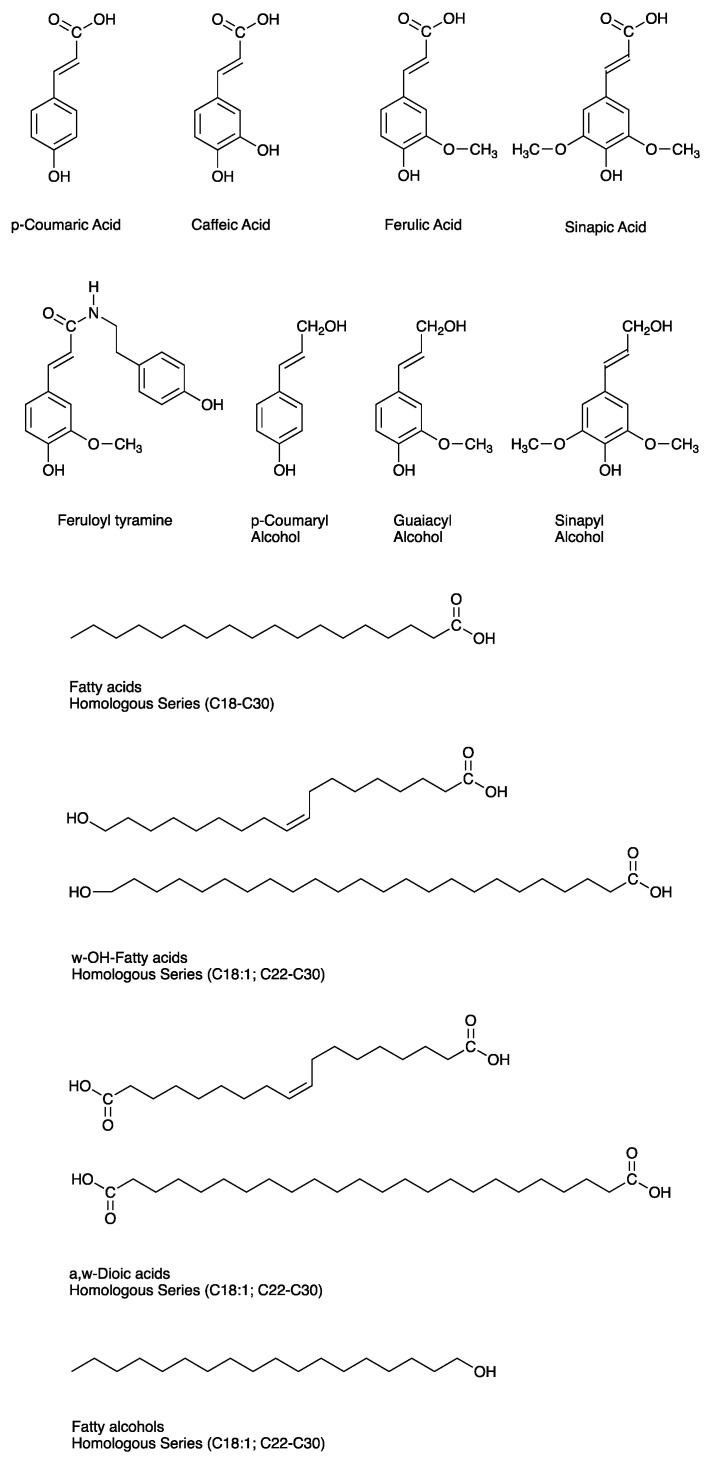
Typical suberin phenolic and aliphatic monomers.

**Figure 3 plants-11-00555-f003:**
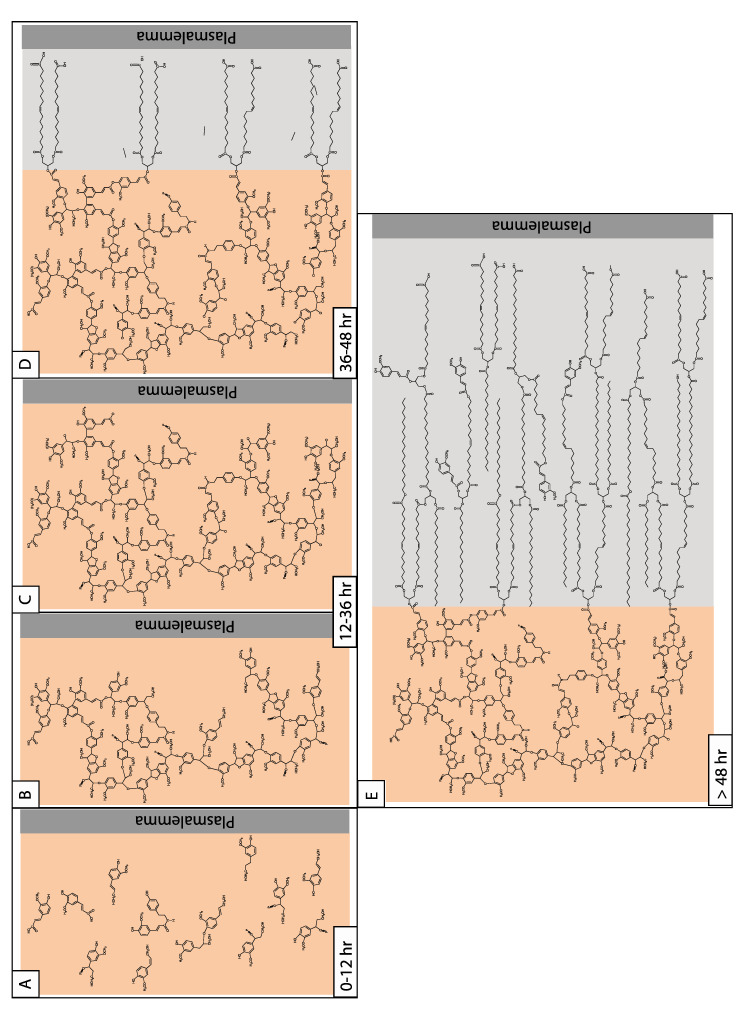
Time course of potato tuber wound suberin assembly. Using gene expression and chemical analysis data form wound healing potato tubers, the temporal deposition of the phenolic and aliphatic monomers of suberin is predicted to begin with phenolics in the cell wall (arrange), followed by aliphatics in the space (grey) between the cell wall and plasmalemma. (**A**) Within 12 hpw, phenolics begin to accumulate in the cell wall and become cross-linked via a peroxidase-mediated process. (**B**,**C**) Between 12-36 hpw, phenolics continue to accumulate and become cross-linked within the cell wall. (**D**) Approx. 36-48 hpw, acyl-hydroxycinnamate esters become cross-linked to the poly(phenolic) matrix, forming a transition between phenolic and aliphatic suberin deposition. (**E**) After 48 hpw, aliphatic suberin monomers/substructures are delivered to the surface of the cell wall where they are cross-linked to aliphatics at the surface of the cell wall.

**Figure 4 plants-11-00555-f004:**
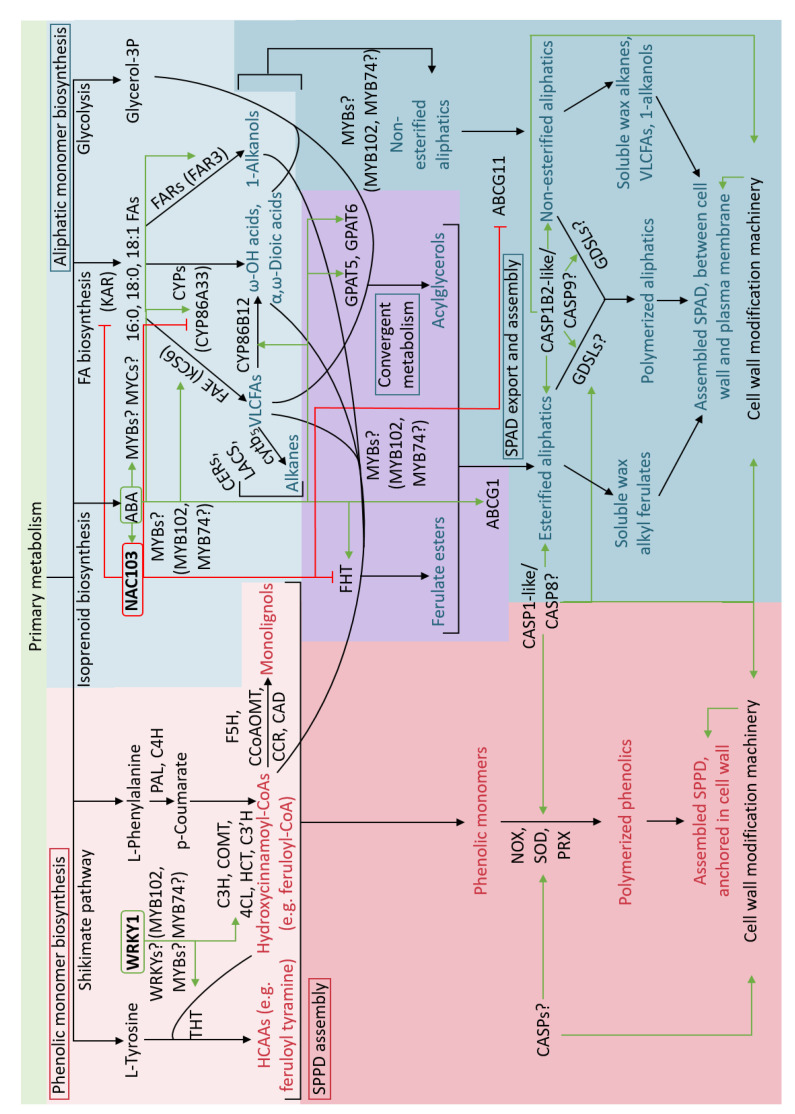
Overview of the regulation of suberin biosynthesis and assembly in wound-healing potato tubers. This overview offers a synthesis of findings, and proposes mechanisms of regulation at the levels of monomer biosynthesis, deposition, polymerization and assembly, based on the literature described in this review. Primary metabolic pathways (shaded green) yield precursors and energy molecules that feed into specialized suberin-related metabolic branches. For example, carbohydrate metabolism yields erythrose-4-phosphate and phospho-*enol*-pyruvate as precursors to the shikimate pathway and production of aromatic amino acids used as precursors for phenolic suberin biosynthesis. Pyruvate and glycolysis-derived glyceraldehyde-3-phosphate are used for the isoprenoid metabolism that yields the phytohormone abscisic acid (ABA) via the carotenoid pathway. Pyruvate is also a substrate for the tricarboxylic acid cycle that yields acetyl-CoA for fatty acid biosynthesis, and results in the generation of 16:0, 18:0 and 18:1 fatty acids that undergo various modifications for aliphatic suberin monomer production. Glycerol-3-phosphate is synthesized from the dehydrogenation of dihydroxyacetone phosphate produced during glycolysis. The biosynthesis of suberin poly(phenolic) domain monomers (dark red) may be regulated by WRKY and MYB TFs. WRKY1 regulates *THT* and *4CL* in relation to phenylpropanoid metabolism in pathogen-infected aerial potato organs. MYB74 and MYB102 may also regulate phenylpropanoid metabolism in potato tuber suberization. ABA regulates the biosynthesis of several key suberin poly(aliphatic) domain monomers (blue) by positively impacting genes involved in their production. NAC103 acts as a transcriptional suppressor of fatty acid and aliphatic suberin-related genes, and is induced by ABA. MYB TFs such as MYB102 and MYB74 may positively regulate aliphatic suberin production. (See Table 1 for key suberin biosynthetic genes and Table 2 for other TFs that regulate phenylpropanoid and/or fatty acid biosynthesis in other species.) It is feasible that potato orthologs of TFs characterized in other species may regulate suberization in wounded tubers. Most phenolic monomers are solely polymerized and incorporated into the SPPD. Feruloyl-CoA can also be conjugated to very long-chain fatty acids (VLCFAs), ω-hydroxy and α,ω-dioic acids, and 1-alkanols to yield ferulate esters, including alkyl ferulates as SPAD-associated soluble wax components. Modified fatty acids can also be esterified to glycerol via the glycerol-3-phosphate acyltransferases (GPATs). Esterified aliphatic constituents are exported by ABCG1. These steps represent a point of convergence between the two major suberin biosynthetic pathways and are labeled as “convergent metabolism” (shaded purple) in the figure. MYB TFs such as MYB102 and MYB74 are putative regulators of the genes encoding convergent metabolic steps. The translocation of aliphatic monomers (alkanes, VLCFAs, modified fatty acids, 1-alkanols) that are not esterified to glycerol or feruloyl-CoA (i.e., soluble components destined for polymerization, or that remain non-polymerized as associated wax) has not been established in potato, but ABCG11 has predicted involvement. Phenolic monomers are thought to undergo a NADPH-dependent oxidase (NOX), superoxide dismutase (SOD), and anionic peroxidase (PRX)-mediated polymerization, whereas SPAD polymerization activities remain uncharacterized. CASP and GDSL-like proteins may play a role in SPAD polymerization. Phenolic suberin associated CASPs may recruit machinery such as NOX, SOD, PRX for polymerization, and influence enzymes involved in the organization of cell wall polysaccharides in a process that may be associated with SPPD deposition via localization of cell wall modifying activities. CASP1-like/CASP8 could regulate the linkage between two domains, their spatial organization, and/or the polymerization and deposition of esterified aliphatics that act as building blocks for SPAD assembly. CASP9 may coordinate aliphatic suberin assembly. GDSLs may act as “suberin synthases” at this stage of assembly, akin to cutin synthases. ABCG1 is required for the export of aliphatic suberin components. *ABCG11* is suppressed by NAC103. ABCGs with varied substrate specificities and MYB TFs are likely involved in the export, deposition and polymerization of non-esterified aliphatics, as well as the organization of non-polymerized, soluble waxes. Cell wall modification machinery may be regulated by CASPs (e.g., CASP1B2-like/CASP9) to organize the deposition of polymerized aliphatics between the cell wall and plasma membrane, prior to secondary cell wall formation. Abbreviations: 4CL, 4-coumarate-CoA ligase; ABA, abscisic acid; ABCG1, ATP-binding cassette (ABC) subfamily G transporter 1; ABCG11, ATP-binding cassette (ABC) subfamily G transporter 11; ABCG6, ATP-binding cassette (ABC) subfamily G transporter 6; C3H, p-coumarate 3-hydroxylase; C3’H, p-coumaroyl quinate/shikimate 3’-hydroxylase; C4H, cinnamic acid 4-hydroxylase; CAD, cinnamyl alcohol dehydrogenase; CASP8, Casparian strip membrane domain protein 8; CASP9, Casparian strip membrane domain protein 9; CCoA***O***MT, caffeoyl-CoA O-methyltransferase; CCR, cinnamoyl CoA reductase; CER, ECERIFERUM; C***O***MT, caffeic acid O-methyltransferase; CYP, cytochrome P450; CYP86A33, cytochrome P450 subfamily 86A 33; CYP86B12, cytochrome P450 subfamily 86B 12; CYTB_5_, cytochrome b_5_; F5H, ferulate 5-hydroxylase; FA, fatty acid; FAE, fatty acid elongase; GDSL, GDSL domain esterase/lipase; Glycerol-3P, glycerol-3-phosphate; GPAT5, glycerol-3-phosphate acyltransferase 5; GPAT6, glycerol-3-phosphate acyltransferase 6; HCAA, hydroxycinnamic acid amide; HCT, hydroxycinnamate transferase; KAR, β-ketoacyl-ACP reductase; KCS6, β-ketoacyl-CoA reductase 6; LACS, long-chain acyl-CoA synthetase; MYB, MYB family transcription factor; NAC, NAC domain transcription factor (NAM, no apical meristem, ATAF, Arabidopsis transcription activation factor, and CUC, cup-shaped cotyledon); NOX, NADPH-dependent oxidase; PAL, phenylalanine ammonia lyase; PRX, anionic peroxidase; SOD, superoxide dismutase; SPAD, suberin poly(aliphatic) domain; SPPD, suberin poly(phenolic) domain; THT, tyramine hydroxycinnamoyl transferase; VLCFA, very-long-chain fatty acid; WRKY, WRKY domain family transcription factor.

**Table 1 plants-11-00555-t001:** Key suberin biosynthetic pathway and assembly steps.

Gene	Corresponding Enzyme Function ^1^	Plant Species ^2^	Reference ^3^
Biosynthesis of phenolic monomers: Phenylpropanoid metabolism			
*PAL*	Phenylalanine ammonia-lyase	Multiple	[52]
*C4H*	Cinnamic acid 4-hydroxylase (C4H)	Multiple	[53]
*4CL*	4-Coumarate-CoA ligase	Multiple	[54]
*HCT*	Hydroxycinnamoyl-CoA transferase	Multiple	[55]
*C3′H*	p-Coumaroyl-quinate-shikimate 3′-hydroxylase	Multiple	[56]
*CCoAOMT*	Caffeoyl-CoA-O-methyltransferase	Multiple	[57]
*F5H*	Ferulate 5-hydroxlyase	Multiple	[58]
*COMT*	Caffeic acid O-methyltransferase	Multiple	[59]
*THT*	Hydroxycinnamoyl-CoA:tyramine N-(hydroxycinnamoyl) transferase	*Solanum tuberosum*	[60]
*CCR*	Cinnamoyl-CoA reductase	Multiple	[61]
*CAD*	Cinnamyl alcohol dehydrogenase	Multiple	[32]
Assembly of the suberin poly(phenolic) domain			
*PRX*	Suberization-associated anionic peroxidase	*Solanum tuberosum*	[62]
*RBOHF*	Respiratory burst oxidase homolog F NADPH oxidases	*Solanum tuberosum, Arabidopsis thaliana*	[63,64,65]
*PER64*	Peroxidase	*Arabidopsis thaliana*	[66]
*ESB1*	Dirigent-like protein	*Arabidopsis thaliana*	[67]
*TPX1*	Cationic peroxidase	*Solanum lycopersicum*	[68,69]
Biosynthesis of aliphatic monomers: Fatty acid elongation, oxidation, and reduction			
*KCS2/DAISY*	β-ketoacyl-CoA synthase	*Arabidopsis thaliana*	[70]
*KCS20*	β-ketoacyl-CoA synthase	*Arabidopsis thaliana*	[12]
*KCS6*	β-ketoacyl-CoA synthase	*Solanum tuberosum*	[10]
*KCR1*	β-ketoacyl-CoA reductase	*Arabidopsis thaliana*	[71]
*PASTICCINO2 (PAS2)*	3-Hydroxyacyl-CoA dehydratase	*Arabidopsis thaliana*	[72]
*ECR*	Enoyl-CoA reductase ^4^	*Arabidopsis thaliana*	[73]
*CYP86A1/HORST*	Cytochrome P450-dependent fatty acid ω-hydroxylase	*Arabidopsis thaliana*	[15,74]
*CYP86B1/RALPH*	Cytochrome P450-dependent fatty acid ω-hydroxylase	*Arabidopsis thaliana*	[16]
*CYP86A33*	Cytochrome P450-dependent fatty acid ω-hydroxylase	*Solanum tuberosum*	[13,14]
*CYP86B12*	Cytochrome P450-dependent fatty acid ω-hydroxylase	*Solanum tuberosum* ^4^	[36]
*NHFAD* ^4^	NADP-dependent ω-hydroxy fatty acid dehydrogenase	*Solanum tuberosum*	[75,76,77]
*NOFAD* ^4^	NADP-dependent ω-oxo fatty acid dehydrogenase	*Solanum tuberosum*	[75,76,77]
*CYP94A5*	Cytochrome P450-dependent fatty acid hydroxylase	*Nicotiana tabacum*	[78]
*CYP94C1*	Cytochrome P450-dependent fatty acid hydroxylase	*Arabidopsis thaliana*	[79]
*FAR1*	Fatty acyl-CoA reductase	*Arabidopsis thaliana*	[18]
*FAR3*	Fatty acyl-CoA reductase	*Solanum tuberosum* ^4^	[36]
*FAR4*	Fatty acyl-CoA reductase	*Arabidopsis thaliana*	[18]
*FAR5*	Fatty acyl-CoA reductase	*Arabidopsis thaliana*	[18]
*ECERIFERUM1 (CER1)*	Very-long-chain aldehyde decarbonylase	*Arabidopsis thaliana*	[80]
*ECERIFERUM3 (CER3)*	Very-long-chain aldehyde decarbonylase	*Arabidopsis thaliana*	[80]
*CYTB5*	Cytochrome b_5_ hemoprotein (cofactor)	*Arabidopsis thaliana*	[80]
*LACS1*	Long-chain acyl-CoA synthase	*Arabidopsis thaliana*	[2]
Esterification, deposition, and assembly of the suberin poly(aliphatic) domain			
*GPAT5*	Glycerol-3-phosphate acyltransferase	*Arabidopsis thaliana, Solanum tuberosum* ^4^	[20,21] [36]
*GPAT6*	Glycerol-3-phosphate acyltransferase	*Solanum tuberosum* ^4^	[36]
*GPAT7*	Glycerol-3-phosphate acyltransferase	*Arabidopsis thaliana*	[81]
*FHT*	Fatty alcohol/fatty ω-hydroxyacid hydroxycinnamoyl acyltransferase	*Solanum tuberosum*	[23,24]
*ASFT/HHT*	Feruloyl transferase/ω-hydroxy acid hydroxycinnamoyltransferase	*Arabidopsis thaliana*	[17,22]
*ABCG1*	ATP-binding cassette subfamily G transporter	*Solanum tuberosum*, *Arabidopsis thaliana*	[25,82]
*ABCG11/WBC11*	ATP-binding cassette subfamily G transporter	*Arabidopsis thaliana*, *Solanum tuberosum*^4^	[82,83]
*ABCG2*	ATP-binding cassette subfamily G transporter	*Arabidopsis thaliana*	[26]
*ABG6*	ATP-binding cassette subfamily G transporter	*Arabidopsis thaliana*	[26]
*ABCG20*	ATP-binding cassette subfamily G transporter	*Arabidopsis thaliana*	[26]
*RCN1/ABCG5*	ATP-binding cassette subfamily G transporter	*Oryza sativa*	[84]
*SUS*	Suberin synthase / GDSL-motif esterase ^4^	Multiple	[85]
*CASP1-like/CASP8*	Casparian strip membrane domain-like protein ^4^	*Solanum tuberosum*	[86]
*CASP1B2-like/CASP9*	Casparian strip membrane domain-like protein ^4^	*Solanum tuberosum*	[86]
*CASPL1B2*	Casparian strip membrane domain-like protein ^4^	*Arabidopsis thaliana*	[85]

^1^ Includes known and predicted functions. ^2^ “Multiple” species reflects general knowledge and/or characterization in 3+ species, e.g., based on known conserved lignin biosynthetic steps. Individual species are listed where applicable. ^3^ Representative references are given. ^4^ Gene and/or enzyme function and involvement in suberization is predicted. Predicted function is based on preliminary characterization (e.g., transcriptomic or proteomic analysis, or identification through experimental observation) and/or putative homology to a counterpart characterized in another species.

## Data Availability

Not applicable.

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
