# Peer review of "Suberin Biosynthesis, Assembly, and Regulation"

_plants, 2022, doi:10.3390/plants11040555_

Round 1

Reviewer 1 Report

The manuscript entitled “Suberin Biosynthesis, Assembly and Regulation” by Woolfson et al. is a very interesting review. It describes the current accepted models (i.e. “polyaliphatic” and “polyaliphatic/polyaromatic” models) for this plant barrier biopolymer, its temporal deposition, the biosynthesis of aromatic and aliphatic monomers and how them can polymerize, the regularization of suberization by phytohormones and transcription factors, and the role for Casparian strip membrane domain proteins. The review is clear, academic, and detailed. In my opinion, it will be very useful for researchers in the topic and other potential readers.

 I have two very minor comments:

1.- I miss a table or figure where main suberin monomers are described. It would also be very useful if the percentages of these molecules for specific tissues/species are included. In my opinion, this can help to understand better suberin complexity.

2.- Recently, Prof. Stark’s team have published a work entitled “Building Blocks of the Protective Suberin Plant Polymer Self-Assemble into Lamellar Structures with Antibacterial Potential” (https://doi.org/10.1021/acsomega.1c04709) about the self-assembly and spontaneous self-esterification of suberin monomers. Is it possible to contextualize these results in the framework of the authors’ review?

Author Response

Reviewer 1

The manuscript entitled “Suberin Biosynthesis, Assembly and Regulation” by Woolfson et al. is a very interesting review. It describes the current accepted models (i.e. “polyaliphatic” and “polyaliphatic/polyaromatic” models) for this plant barrier biopolymer, its temporal deposition, the biosynthesis of aromatic and aliphatic monomers and how them can polymerize, the regularization of suberization by phytohormones and transcription factors, and the role for Casparian strip membrane domain proteins. The review is clear, academic, and detailed. In my opinion, it will be very useful for researchers in the topic and other potential readers.

 Response: We thank the reviewer for their positive and supportive comments.

 I have two very minor comments: 

1.- I miss a table or figure where main suberin monomers are described. It would also be very useful if the percentages of these molecules for specific tissues/species are included. In my opinion, this can help to understand better suberin complexity.

Response: We have added a new figure (new Figure 2) that shows the structures of the major monomers of the suberin macromolecule. We stopped short of summarizing the percentages of these across species, and instead direct readers to specific citations where the composition of suberin monomers is detailed.

2.- Recently, Prof. Stark’s team have published a work entitled “Building Blocks of the Protective Suberin Plant Polymer Self-Assemble into Lamellar Structures with Antibacterial Potential” (https://doi.org/10.1021/acsomega.1c04709) about the self-assembly and spontaneous self-esterification of suberin monomers. Is it possible to contextualize these results in the framework of the authors’ review?

Response: We have incorporated the recent work by Ruth Stark’s group into Section 3.3.3. Possible mechanisms for aliphatic monomer polymerization.

Reviewer 2 Report

The review by Woolfson et al. covers all known aspects of suberin biosynthesis and deposition. The manuscript is very dense and complement nicely a recent review of Shukla and Barberon (2021). It is very informative and reflect the hudge complexity of this polymer.

My main comment is the lack of a figure showing the (different) current model for suberin structure and integration into the cell wall. In addition, some tables compiling the major genes in the different steps described would also be useful for the reader.

There are some minor mistakes such as gene or species names that are not always in italics (for instance line 683, 710, 712, 715.

Line 719: SUBERMAN coding for a MYB ... or remove the italics

Italics in the reference.

In table 1, what do you mean by missing citation, see comment?

The O- of CCoAOMT and COMT should be in italics.

Author Response

Reviewer 2

The review by Woolfson et al. covers all known aspects of suberin biosynthesis and deposition. The manuscript is very dense and complement nicely a recent review of Shukla and Barberon (2021). It is very informative and reflect the hudge complexity of this polymer.

 Response: We thank the reviewer for their positive and supportive comments.

My main comment is the lack of a figure showing the (different) current model for suberin structure and integration into the cell wall. In addition, some tables compiling the major genes in the different steps described would also be useful for the reader.

Response: We have added a new figure (new Figure 1) comparing the two main models of suberin macromolecular structure. One model, referred to as an “integrated model” is based on Kolattukudy’s original model proposed in the early 1980s and revised by Jose Graca in 2015. The second is a “two-domain model” that we proposed in the early 2000s. We hope that this figure (and a revised description in the caption and text) capture what the reviewer felt was lacking.

We also added a table (new Table 1) that summarizes the genes involved in suberin biosynthesis. We were unclear exactly what the reviewer wanted. Consequently, we have avoided preparing a comprehensive compilation of gene IDs, and instead provide a summary of the genes described in the text, listed in order of biosynthetic relevance. We hope this meets with the reviewer’s expectations.

Lastly, we have addressed all of the minor issues noted by the reviewer, and have gone through the manuscript to carefully edit any further instances where italics are required, taking care to scrutinize each instance of gene/protein mention. We have also corrected some language to minimize ambiguities.